# Converting organosulfur compounds to inorganic polysulfides against resistant bacterial infections

Zhuobin Xu[1], Zhiyue Qiu[1], Qi Liu[1], Yixin Huang[1], Dandan Li[1], Xinggui Shen[2], Kelong Fan[3], Juqun Xi[1], Yunhao Gu[1], Yan Tang[1], Jing Jiang[1], Jialei Xu[4], Jinzhi He[4], Xingfa Gao[5], Yuan Liu[6], Hyun Koo[6], Xiyun Yan[3] & Lizeng Gao [1,7]

The use of natural substance to ward off microbial infections has a long history. However, the large-scale production of natural extracts often reduces antibacterial potency, thus limiting practical applications. Here we present a strategy for converting natural organosulfur compounds into nano-iron sulfides that exhibit enhanced antibacterial activity. We show that compared to garlic-derived organosulfur compounds nano-iron sulfides exhibit an over 500-fold increase in antibacterial efficacy to kill several pathogenic and drug-resistant bacteria. Furthermore, our analysis reveals that hydrogen polysulfanes released from nano-iron sulfides possess potent bactericidal activity and the release of polysulfanes can be accelerated by the enzyme-like activity of nano-iron sulfides. Finally, we demonstrate that topical applications of nano-iron sulfides can effectively disrupt pathogenic biofilms on human teeth and accelerate infected-wound healing. Together, our approach to convert organosulfur compounds into inorganic polysulfides potentially provides an antibacterial alternative to combat bacterial infections.

[1] Institute of Translational Medicine, Department of Pharmacology, School of Medicine, Yangzhou University, Yangzhou, Jiangsu, 225001, China. [2] Pathology and Translational Pathobiology, Louisiana State University Health Sciences Center, Shreveport, LA, 71130-3932, USA. [3] Key Laboratory of Protein and Peptide Pharmaceuticals, Institute of Biophysics, Chinese Academy of Sciences, Beijing, 100101, China. [4] State Key Laboratory of Oral Diseases, Department of Endodontics, West China Hospital of Stomatoloty, Sichuan University, Chengdu, Sichuan, 610041, China. [5] College of Chemistry and Chemical Engineering, Jiangxi Normal University, Nanchang, Jiangxi, 330022, China. [6] Biofilm Research Labs, Levy Center for Oral Health, School of Dental Medicine, University of Pennsylvania, Philadelphia, PA, 19104, USA. [7] Jiangsu Key Laboratory of Experimental and Translational Non-coding RNA Research, Yangzhou University, Yangzhou, Jiangsu, 225001, China. These authors contributed equally: Zhuobin Xu, Zhiyue Qiu, Qi Liu. Correspondence and requests for materials should be addressed to L.G. (email: lzgao@yzu.edu.cn)

Natural substances have been used to prevent and treat microbial diseases since ancient times. For instance, the use of bulbs of plants in *Allium* genus, such as garlic and onion for their antimicrobial activities has been described as early as the 16th century BC[1–3]. Like the traditional Welsh rhyme which says, "Eat leeks in March and wild garlic in May, and all the year after the physicians may play"[4]. Currently, most medicinal garlic products are manufactured from the extracts (e.g., garlic oil) which contain organosulfur compounds[5–7]. These organosulfur compounds contain one or more sulfur atoms bonded with carbon, which is the basis for their biological activities, including antimicrobial, antioxidant, antitumor and antiasthmatic activities[8].

The main antibacterial component present in garlic bulbs is allicin[9,10], which is converted by the enzyme alliinase from alliin (a derivative of amino acid cysteine)[11]. Allicin is unstable under physiological conditions and might quickly transform into alkyl sulfides, such as diallyl disulfide (DADS), diallyl trisulfide (DATS) and diallyl sulfide (DAS) (Fig. 1a). Transformed products are assumed to be the active ingredients in plants responsible for suppressing bacterial growth. However, these purified components often exhibit weak antibacterial effects, while a small number of these components are devoid of any bioactivity[12,13]. In addition, many organosulfur compounds are volatile and thus exhibit unpleasant odors. Furthermore, they are characterized by poor water solubility, further limiting wider biomedical and clinical applications. However, recent studies have demonstrated that inorganic sulfur in the form of nanoscale metal sulfides, such as copper sulfide (CuS) and molybdenum disulfide ($MoS_2$), exhibits high antibacterial activity[14,15]. Thus, converting organosulfur compounds into nanoscale inorganic sulfides represents a promising route to improve the efficacy of existing antibacterial products.

Bacterial infections pose a considerable threat to public health as many pathogenic bacteria readily develop resistance to multiple antibiotics and form biofilms with additional protection from antibiotic treatment[16]. Therefore, it is urgently required to develop new antibacterial agents[17]. Here we develop an approach to convert natural organosulfur compounds into nanometer-sized iron sulfides (nFeS). We find that compared with organosulfur compounds, the nFeS exhibits enhanced antibacterial activity to pathogenic bacteria with drug resistance or associated with biofilms.

## Results

### Conversion of natural organosulfur compounds into nFeS.
To convert organosulfur compounds into inorganic sulfides, we employed a solvothermal method that produces inorganic cystals in a sealed autoclave operated at high temperature and high vapor pressure. We previously used this method to synthesize ferromagnetic nanocrystals ($Fe_3O_4$) by reducing ferric iron ($Fe^{3+}$) into $Fe_3O_4$ nanocrystal in ethylene glycol (EG) solvent containing sodium acetate (NaOAc) at 200 °C[18]. The presence of an elemental sulfur in this reaction preferentially causes the formation of iron sulfide[19]. As shown in Fig. 1b, we directly introduced the natural organosulfur compound into the solvothermal reaction and found that sheet-like and hexagonal nanostructures were formed rather than the typical nanoparticles for $Fe_3O_4$. The hexagonal plates exhibited a length of up to 1 µm and a thickness of up to 20–30 nm. Using energy dispersive spectrometry (EDS), we found that these nanostructures consisted of iron and sulfide (Supplementary Fig. 1a). X-ray diffraction (XRD) showed that pyrrhotite ($Fe_{1-x}S$, 29-0724) and gregite ($Fe_3S_4$, 16-0713), two phases of iron sulfides were present in the product (Fig. 1c). High-resolution transmission electron microscope (HRTEM) images

taken from the large irregular thin sheet exhibited clear lattice fringes with a d-spacing of 0.298 nm. In addition, the selected-area electron diffraction (SAED) patterns showed that the nanosheets displayed a single crystalline structure with (311) lattice planes of $Fe_3S_4$. In comparison, HRTEM images taken from the hexagonal plate showed that lattice fringes were formed, with a d-spacing at 0.15 nm, which can be assigned to the (400) plane of $Fe_{1-x}S$. However, the SAED patterns taken from the edge of the hexagonal plate showed that $Fe_{1-x}S$ was polycrystalline (Fig. 1d and Supplementary Fig. 1b). We observed these features in the products generated from the additive organosulfur compounds including DADS, DATS, cysteine, and its derivatives, such as cystine and glutathione (GSH). The resulting products were denoted DADS-nFeS, DATS-nFeS, Cys-nFeS, Cyss-nFeS and GSH-nFeS (Supplementary Fig. 2). However, diallyl sulfide (DAS) showed negligible nFeS formation; also, allyl methyl sulfide (AMS) and methionine failed to form any nFeS in the solvothermal reaction (Supplementary Fig. 2 and Supplementary Fig. 3). The successful synthesis of nFeS by solvothermal reduction of $Fe^{3+}$ with cysteine and GSH was presumably achieved due to the presence of highly chemically reductive thiol groups in these two sulfides. The high conversion rate of di- and tri-sulfide into nFeS is presumably due to their relative weak covalent bonds connecting their S atoms. As shown in Fig. 1b, the calculated values of bond dissociation energy (BDE) for C–S in DADS, DATS, and cystine were 35.32, 38.68, and 51.89 kcal mol$^{-1}$, respectively, which were smaller than those in DAS (69.358/50.69 kcal mol$^{-1}$) and methionine (66.90/70.69 kcal mol$^{-1}$). The low BDE of C–S may favor thermal degradation to release sulfur which further reacts with iron to form nFeS in the solvothermal reaction. These results indicate that two reactions may occur in the synthetic process: sulfur extraction from organosulfur compounds via pyrolysis, followed by the reaction of nFeS formation (Fig. 1e).

We then investigated the correlation between nFeS formation and the amount of added organosulfur compound. We chose cysteine as a standard molecule because it is a general precursor for many natural organosulfur compounds. In addition, compared to garlic-derived organosulfides (e.g., DADS or DATS) (Supplementary Fig. 4), cysteine is odorless, soluble in many solvents, thus it is more suitable for both solvothermal conversion and antibacterial testing. Cysteine was added to the solvothermal reaction (total volume was 50 mL) in various amounts of 0.1, 0.25, 0.5, 0.75, and 1.0 g (the products are denoted $Cys_{0.1}$-nFeS, $Cys_{0.25}$-nFeS, $Cys_{0.5}$-nFeS, $Cys_{0.75}$-nFeS, and $Cys_{1.0}$-nFeS, respectively). As shown in Supplementary Fig. 5, the formation of nFeS was dependent on the amount of cysteine added to the solvothermal reaction. Consequently, the atomic ratio of sulfide to iron increased from $Cys_{0.1}$-nFeS to $Cys_{1.0}$-nFeS, in which the ratio of Fe/S was up to 48.75/51.25, while that for oxygen to iron decreased correspondingly (Supplementary Fig. 6a and Supplementary Table 1). The $Cys_{1.0}$-nFeS product only contained nanosheets and hexagonal structures whereas $Fe_3O_4$ nanoparticles were present in the $Cys_{0.5}$-nFeS product. Simultaneously, with increasing concentrations of cysteine in the reaction, the degree of magnetism in the nFeS products decreased. The $Cys_{1.0}$-nFeS product exhibited much lower levels of magnetism compared to the $Cys_{0.5}$-nFeS product (Supplementary Fig. 6b). It is well established that while the metastable $Fe_3S_4$ phase exhibits paramagnetism, the stable $Fe_{1-x}S$ phase fails to exhibit magnetism. Thus, the formation of $Fe_{1-x}S$ may be dominant when sufficient amounts of organosulfur are present in the solvothermal reaction (Fig. 1e). Taken together, adding natural organosulfur compounds to the solvothermal process led to the formation of nFeS, and the type and amount of added organosulfur compounds determined the crystalline phase and nanostructure of nFeS products.

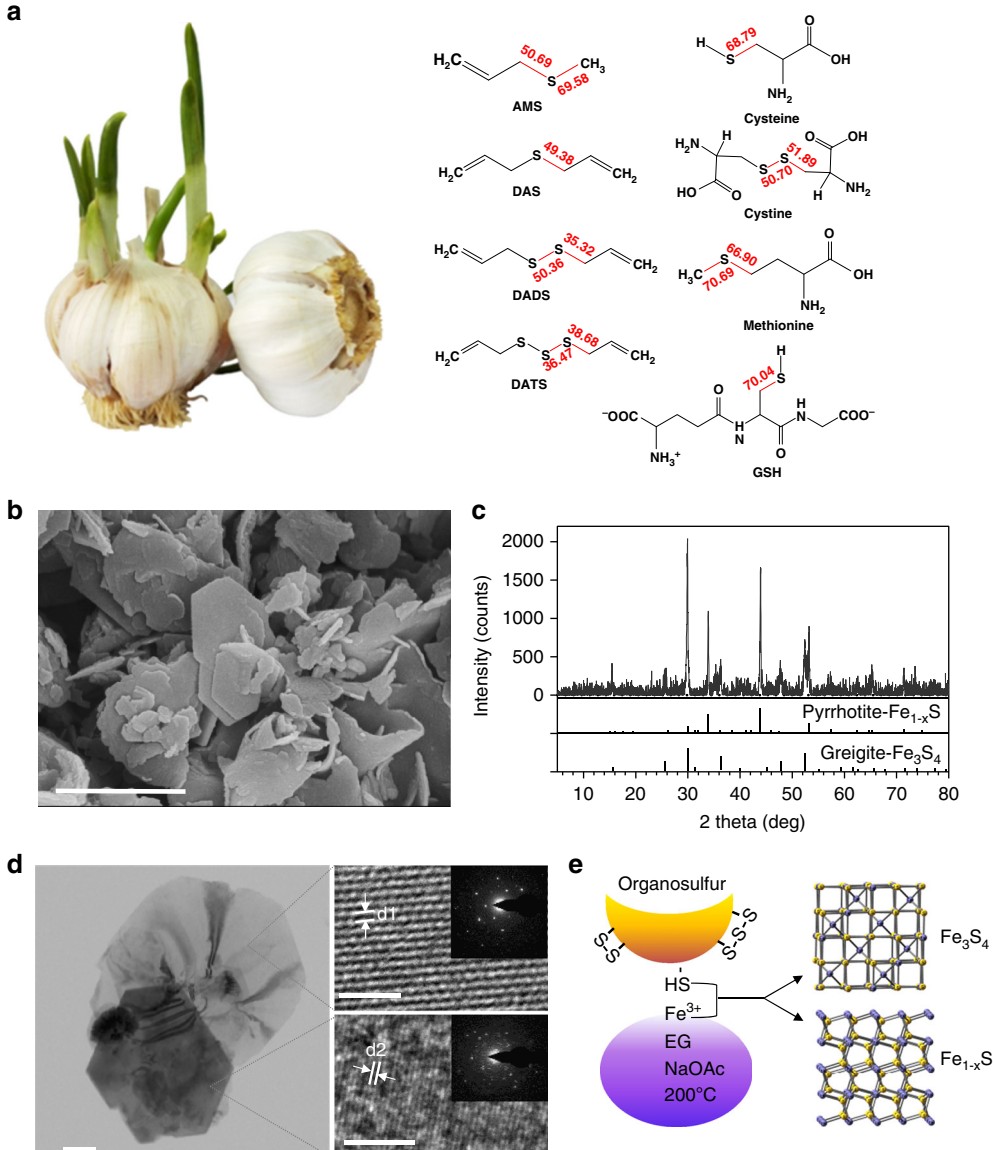

**Fig. 1** Converting organosulfur compounds into nano-iron sulfide (nFeS) by solvothermal synthesis. **a** Garlic-derived and other natural organosulfur compounds. The number in red is the computational calculated value of bond dissociation energy (BDE) for the S-related bonds (in kcal mol$^{-1}$). **b** The scanning electron microscope (SEM) image of nFeS with sheet-like hexagonal nanostructure. Scale bar: 1 μm. **c** XRD characterization of nFeS in the two phases of Fe$_{1-x}$S and Fe$_3$S$_4$. **d** The transmission electron microscope (TEM) image (left image) of nFeS with high resolution characterization (right images) and single-crystal diffraction (inserted images). Left scale bars: 200 nm. Right scale bars: 2 nm. d1: d-spacing at 0.298 nm. d2: d-spacing at 0.15 nm. **e** Scheme of converting organosulfur compounds to nFeS with Fe$_3$S$_4$ and Fe$_{1-x}$S. All experiments were performed in triplicate, and the representative images are shown

**Antibacterial properties of nFeS**. In order to evaluate the antibacterial activity of our nFeS products, we performed quantitative analysis on bacterial viability using colony-forming unit (CFU) method. As shown in Fig. 2a, nFeS produced from different organosulfur compounds exhibited similar antibacterial activity against *Streptococcus mutans* (*S. mutans*), a virulent oral pathogen and well-characterized biofilm-forming organism. In comparison, iron oxide (Fe$_3$O$_4$) nanoparticles failed to exert antibacterial activity, while the corresponding organosulfur compounds exhibited negligible antibacterial activity (Fig. 2b). For instance, DADS-nFeS showed a more than 500-fold increase in bacterial killing (*S. mutans*) activity compared with pure DADS (normalized to the sulfur amount used in DADS-nFeS) (Supplementary Fig. 7). In particular, nFeS derived from cysteine demonstrated dose-dependent

antibacterial activity based on the cysteine amount, whereby Cys$_{0.5}$-nFeS showed the highest biocidal activity (Fig. 2c). Cys$_{0.5}$-nFeS at 0.5 mg mL$^{-1}$ was able to kill bacteria with 3-log reduction of viability (from $10^7$ to $10^4$ CFU mL$^{-1}$) within 10 min of treatment (Supplementary Fig. 8). In addition, the antibacterial efficacy was dependent on the dosage and time. Our antibacterial tests also showed that Cys$_{0.5}$-nFeS displayed a broad antibacterial activity against Gram-negative bacteria, including *Escherichia coli* (*E. coli*), *Pseudomonas aeruginosa* (*P. aeruginosa*), *Salmonella enteritidis* (*S. enteritidis*) and Gram-positive bacteria such as *S. mutans*, *Staphylococcus aureus* (*S. aureus*) as well as drug-resistant strains of *S. aureus* (Fig. 2d–h and Supplementary Fig. 9).

To further assess the antibacterial mechanism of nFeS, we investigated the oxidative state and impact on both bacterial

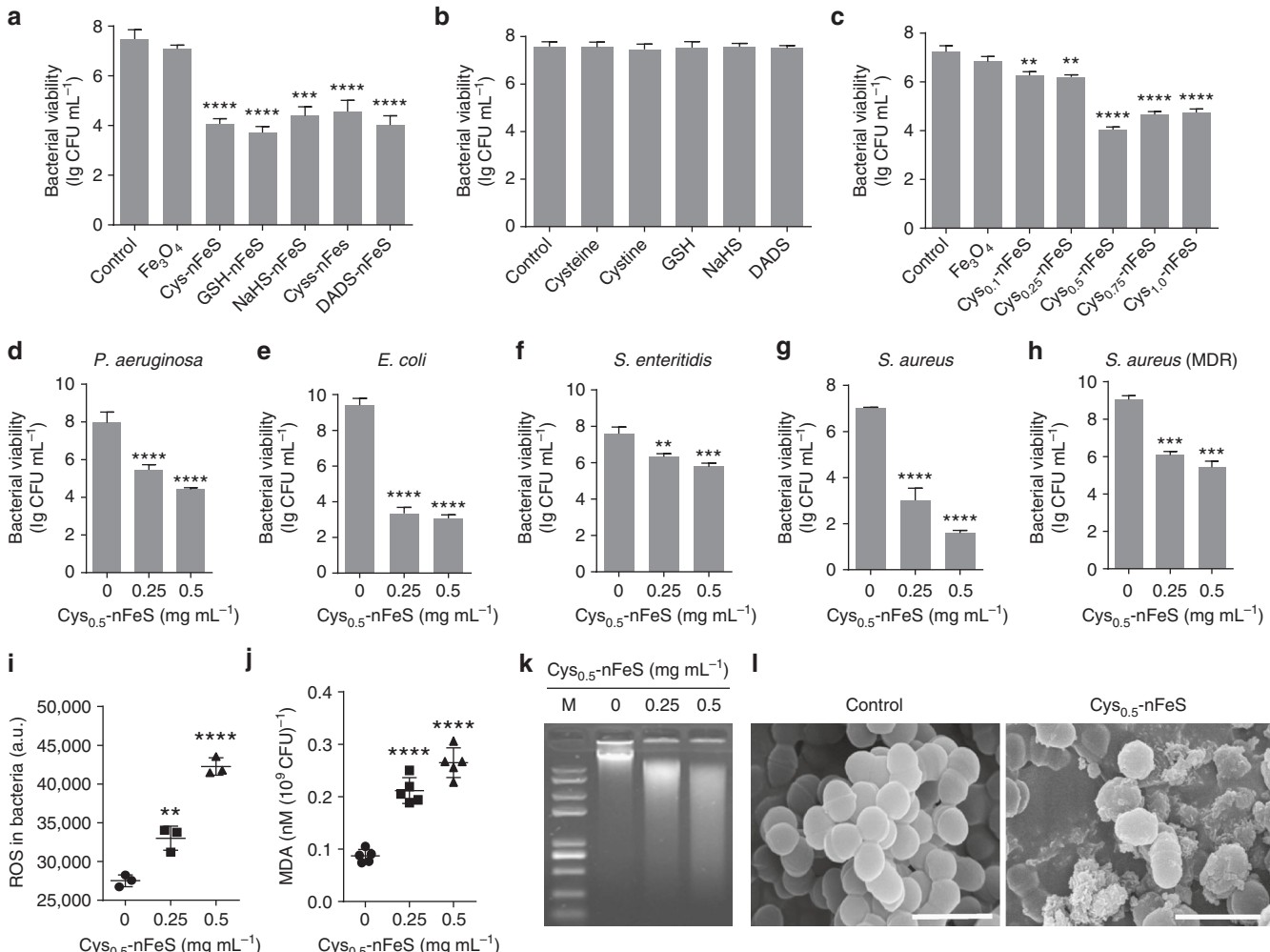

**Fig. 2** Antibacterial activity of nFeS. **a** Antibacterial activity of nFeS converted from different organosulfur sources against *S. mutans* UA159 (a biofilm-forming dental pathogen). Cys-nFeS was derived from cysteine with 0.5 g additive in solvothermal conversion, other nFeS products were synthesized with organosulfur compounds at the amount normalized to the same sulfur content in 0.5 g cysteine. The concentration for each nFeS was adjusted to 0.5 mg mL$^{-1}$ in all antibacterial tests. **b** Antibacterial (*S.mutans* UA159) activity of the organosulfur compounds. The concentration for each sulfur compound was normalized to sulfur amount equal to that contained in 0.5 mg mL$^{-1}$ of Cys$_{0.5}$-nFeS. **c** Dependence of antibacterial (*S. mutans* UA159) efficacy of Cys-nFeS on the amount of cysteine input to the solvothermal synthesis. The concentration for Fe$_3$O$_4$ nanoparticle or nFeS was at 0.5 mg mL$^{-1}$. **d–h** Antibacterial activity on *P. aeruginosa*, *E. coli*, *S. enteritidis*, *S. aureus*, and *S. aureus* (MDR), respectively. **i, j** ROS level and lipid peroxidation of bacteria treated by Cys-nFeS. **k** Genomic DNA degradation of bacteria treated by Cys-nFeS. M: DNA marker. **l** SEM image of bacteria treated by Cys-nFeS. Scale bars: 1 μm. Data are shown as the mean ± s.d. Statistical significance was assessed by unpaired Student's two-sided *t*-test compared to the control group. **$p < 0.01$, ***$p < 0.001$ and ****$p < 0.0001$. Mean values and error bars were defined as mean and s.d., respectively. All experiments were performed in triplicate, and representative images are shown

surfaces and intracellular molecules. The level of reactive oxygen species (ROS) increased by >50%, while lipid peroxidation as measured using a MDA assay increased threefold in the treated *S. mutans* (Fig. 2i, j). In addition, DNA degradation was triggered following the treatment with Cys$_{0.5}$-nFeS (Fig. 2k). Characterization of bacterial surfaces by SEM showed that the morphology and structural integrity of the treated bacteria were also severely affected (Fig. 2l). Similar to Cys-nFeS, DADS-nFeS exhibited antibacterial features, whereas DADS performed poorly (Supplementary Figs. 10, 11). Together, these results strongly suggest that our nFeS possesses effective antibacterial activity against pathogenic and resistant bacteria.

**nFeS releases polysulfanes for antibacterial activity**. To determine the active component in nFeS product, we analyzed the physicochemical changes our nFeS underwent during the antibacterial process. As shown in Fig. 2l, small grain-like

nanoparticles, rather than nanosheets or hexagonal plates, surrounded the bacteria following their treatment with Cys$_{0.5}$-nFeS. This observation indicates that nFeS is structurally transformed during incubation with bacterial cells. To verify this assumption, Cys$_{0.5}$-nFeS was directly dissolved in water, and the physical characteristics and nanostructural changes were analyzed over time. As shown in Fig. 3a, the solution containing Cys$_{0.5}$-nFeS gradually changed color from black to orange with increasing incubation time. Concomitantly, the structure of Cys$_{0.5}$-nFeS transformed from flakes into nanoparticles (Fig. 3b). The same color change and nanostructure transformation were also observed in the DADS-nFeS solution, indicating that this phenomenon is common for nFeS (Supplementary Fig. 12). Furthermore, the typical sheet structure disappeared, together with the hexagonal plates. Also, grain-like nanoparticles were formed after 96 h of incubation. The collected nanoparticles only contained iron and oxygen, as identified by EDS (Supplementary

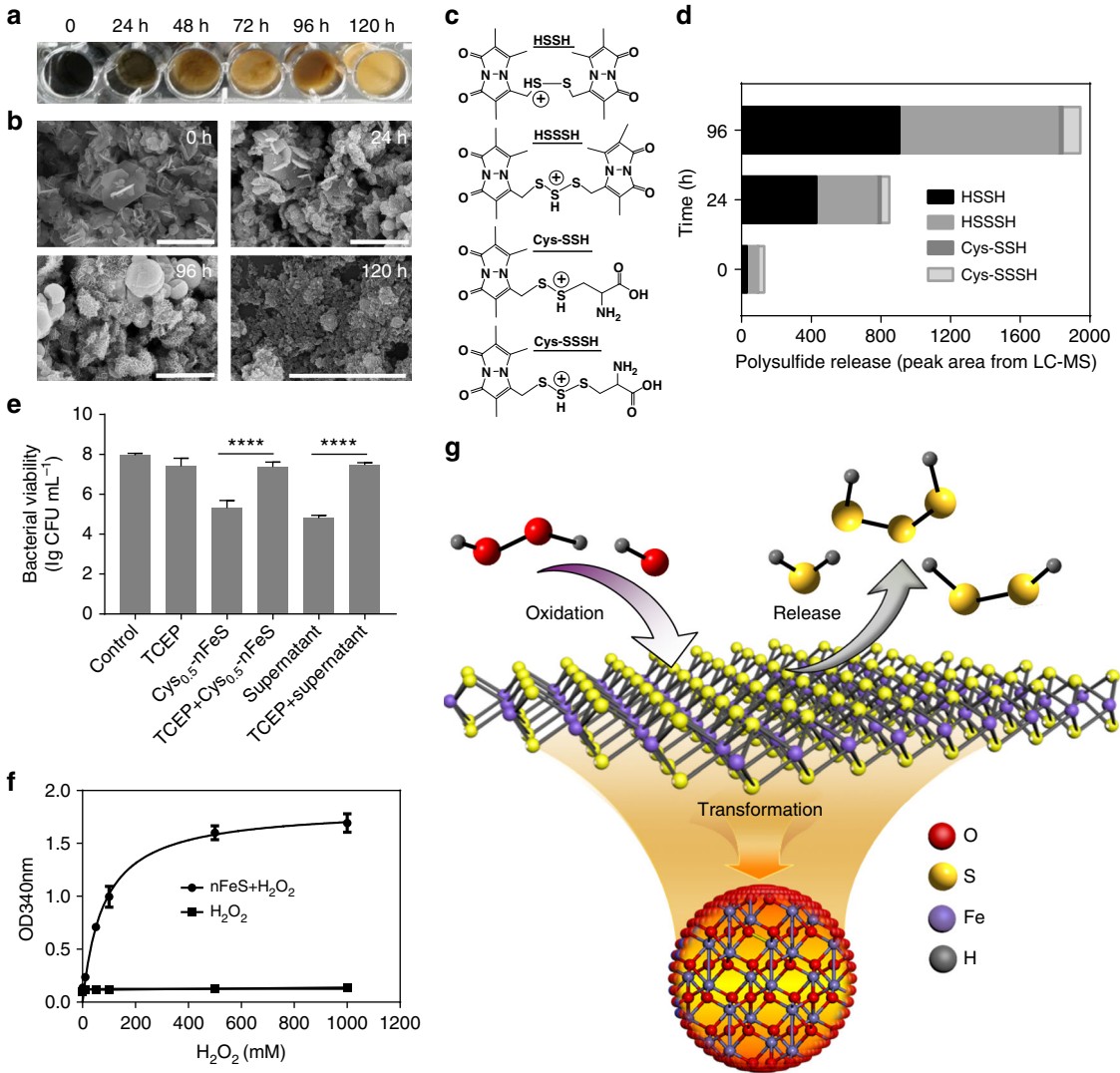

**Fig. 3** nFeS releases polysulfanes for antibacterial activity. **a** Color change of Cys-nFeS incubated in water for different time periods. **b** SEM image of the nanostructure transformation of Cys-nFeS after incubation in water. Scale bars: 2 μm. **c** Fragmentation scheme for polysulfanes identified from the supernatant after monobromobimane (MBB) derivatization in positive ionization mode. **d** Comparison of component ratio of polysulfanes in the supernatant by LC–MS/MS. HSSH: $H_2S_2$. HSSSH: $H_2S_3$. **e** Blocking of antibacterial activity by tris(2-carboxyethyl)phosphine (TCEP). **f** $H_2O_2$ accelerated polysulfane release via catalysis-accelerated release (CAR) effect. **g** Scheme of polysulfane release from nFeS. Data are shown as the mean ± s.d. Statistical significance was assessed using unpaired Student's two-sided $t$-test compared to the control group. $**p < 0.01$, $***p < 0.001$ and $****p < 0.0001$. Mean values and error bars were defined as mean and s.d., respectively. All experiments were performed in triplicate, and representative images are shown

Fig. 13), thus indicating that the color change may be due to the formation of iron oxide ($Fe_2O_3$).

However, iron oxide is not an effective antibacterial agent, thus other components might be responsible for the observed bactericidal activity. To address this question, we next analyzed the chemical species of the sulfide product formed following the nanostructure transformation. We identified the presence of sulfur components in the supernatant of the $Cys_{0.5}$-nFeS solution using liquid chromatographic-tandem mass spectrometric (LC–MS/MS). Monobromobimane (MBB) was introduced into the supernatant to form stable sulfur derivatives, thus allowing evaluation of sulfur species by LC–MS/MS[20,21] (Supplementary Fig. 14). As shown in Fig. 3c, we identified free inorganic sulfides, including hydrogen sulfide ($H_2S$), hydrogen disulfane ($H_2S_2$) and hydrogen trisulfane ($H_2S_3$) in the supernatant. In particular, $H_2S_2$ and $H_2S_3$ were dominant among the identified polysulfane products (Fig. 3d, Supplementary Table 2 and Supplementary Fig. 15). These results indicate that during nanostructure

transformation, iron oxide nanoparticles were formed and free sulfides were released, which is a process of iron sulfide oxidation in the aqueous solution[22]. Thus, our results strongly suggest that nFeS is a donor to release free sulfides in aqueous condition.

We speculated that this release of sulfides provides key factors associated with the antibacterial activity of nFeS, as they are active to interact with glutathione, proteins and enzymes in a living cell. Thus, we first assessed the antibacterial effects of both supernatant and precipitate of $Cys_{0.5}$-nFeS. As shown in Fig. 3e, the supernatant inhibited *S. mutans* growth at levels similar to those measured for the original mixture (with both precipitate and supernatant); whereas, the precipitate itself did not exert biocidal effects, indicating that the antibacterial activity was provided by an active component present in the supernatant. As several types of free sulfides are present in the supernatant, it was necessary to identify the type of sulfide responsible for the observed antibacterial activities. The result shown in Fig. 2b provided initial evidence that sodium hydrosulfide (NaHS), a typical $H_2S$

donor, lacked the antibacterial activity. Thus, we excluded $H_2S$ as the possible antibacterial agent. Previously, the antibacterial activity of garlic oil had been associated with the presence of organosulfur compounds, especially in the form of polysulfides. The antibacterial activity might therefore have been derived from hydrogen polysulfanes (such as $H_2S_2$ and $H_2S_3$). To confirm this hypothesis, polysulfanes in the supernatant of $Cys_{0.5}$-nFeS were eliminated by reducing disulfide bonds in the presence of tris(2-carboxyethyl)phosphine (TCEP). As shown in Fig. 3e, the antibacterial effects disappeared when TCEP was added to both the original mixture as well as into the supernatant of nFeS. This finding provides direct evidence that the polysulfanes released from nFeS are critical factors in reducing bacterial viability.

Interestingly, nFeS also exhibited enzyme-mimicking properties to boost the release of polysulfanes. We speculated that nFeS may be a nanozyme with peroxidase-like and catalase-like activities, which decomposes $H_2O_2$ into free radicals and oxygen, respectively[23–25]. We found that compared to $Fe_3O_4$ nanoparticles, nFeS possesses enhanced peroxidase-like and catalase-like activities (Supplementary Figs. 16, 17, Supplementary Tables 3 and 4). More specifically, Cys0.5-nFeS performed the highest specific activity which was almost 10-fold higher than that of $Fe_3O_4$ nanoparticles (Supplementary Table 5). Importantly, we found that during the enzyme-like catalytic process, the color of the nFeS solution ($1.0 \, mg \, mL^{-1}$) became orange within 30 min in the presence of 50 mM $H_2O_2$ (Supplementary Fig. 18), which was much faster than the color change shown in Fig. 3a. To quantitatively measure the correlation between sulfide release and $H_2O_2$ concentration, the absorbance spectrum was scanned using NaHS as a standard donor of free sulfides, and 340 nm was chosen as the specific absorbance ($A_{340}$) for free sulfide in aqueous solution (Supplementary Fig. 19a). When we measured absorbance for the nFeS supernatant, we found that its absorbance spectrum was similar to that of NaHS (Supplementary Fig. 19b). Therefore, we chose $A_{340}$ to characterize free sulfides release from nFeS. In addition, the value of $A_{340}$ from the $Cys_{0.5}$-nFeS ($1 \, mg \, mL^{-1}$) supernatant reached a plateau in the first 30 min, indicating that the release of sulfide is fairly fast (Supplementary Fig. 19c). The correlation curve between $A_{340}$ and the $H_2O_2$ concentration showed that the release kinetics was similar to those in enzyme-like activities of nFeS, in which 80.25 mM $H_2O_2$ led to the release of half the amount of maximum free sulfides, indicating that free sulfide release was driven by catalysis (Fig. 3f). Following the release of free sulfides, the precipitate from the nFeS exhibited low enzyme-like activity (Supplementary Fig. 20), which is consistent with the catalytic performance of iron oxide. Based on these results, we deduced that in the presence of $H_2O_2$, oxidation may quickly occur on the surface of nFeS, leading to accelerated release of free sulfides.

On the basis of these observations, we propose that the mechanism for the antibacterial activity of nFeS involves the release of polysulfanes during the oxidation of nFeS to iron oxide (Fig. 3g). In addition to Cys-nFeS, nFeS products derived from other organosulfur compounds, such as DADS, also possess these enzyme-like activities (Supplementary Fig. 21) and thus are able to quickly release polysulfanes when exposed to $H_2O_2$. As the release is enhanced by the intrinsic enzyme-like activities of nFeS, we named this phenomenon catalysis-accelerated release (CAR). Taken together, these results indicate that the antibacterial activity of nFeS originates from hydrogen polysulfanes released in the process of nanostructure transformation. The release might be improved by the addition of $H_2O_2$, which is ascribed to the CAR effect of nFeS.

**S. mutans biofilm elimination using nFeS.** The unique multi-functional properties and enhanced antibacterial activity suggest

that nFeS is a suitable agent for effective therapeutic approaches that prevent or treat bacterial infections. To evaluate the ability of nFeS for cariogenic biofilm elimination, we used two distinct infection models and topical treatment regimen akin of clinical situation. First, we examined whether nFeS can disrupt the capacity of cariogenic pathogen S. mutans to form biofilms on saliva-coated hydroxyapatite (sHA) surface, a tooth enamel-like material commonly used for dental biofilm experiments (Supplementary Fig. 22a, b)[17,26–28]. To simulate topical exposure, $Cys_{0.5}$-nFeS was applied three times (0, 19 and 29 h, 10 min time$^{-1}$) by immersing the sHA discs into $Cys_{0.5}$-nFeS solutions during the course of biofilm growth. As shown in Supplementary Fig. 23a, b, we found that biofilm biomass was markedly disrupted (dry weight, 90% reduction), with bacteria viability considerably reduced (~6-log reduction of viable bacteria) in nFeS-treated biofilms, demonstrating that nFeS effectively disrupted S. mutans biofilms. In contrast, neither commercial bulk iron sulfide (bFeS) nor cysteine exhibited any detectable anti-biofilm activity.

In addition, SEM imaging showed that when topically treated with $Cys_{0.5}$-nFeS, no biofilm structure was formed on the sHA surface (Supplementary Fig. 23c). Importantly, while DADS alone failed to reduce the biofilm biomass and cell viability, DADS-nFeS successfully prevented biofilm formation (Supplementary Fig. 24). Together, these results demonstrate that nFeS is the principle agent for biofilm disruption. In addition, the nFeS was able to kill bacteria embedded in a biofilm matrix, indicating that the released polysulfanes can permeate into bacterial biofilm.

To evaluate the impact of nFeS treatment on biofilm integrity, the 3D structure of the biofilm was analyzed by staining the bacterial cells with Syto 9 (Green) and labeling the extracellular polymeric matrix with Alexa Fluor 647 (Red) prior to confocal laser scanning microscopy (CLSM) analysis (Supplementary Fig. 25)[29]. As shown in Fig. 4a, only small and sparsely distributed bacterial clusters and negligible extracellular matrix were observed following nFeS topical treatment, resulting in highly disrupted biofilm structure. These findings clearly demonstrate that nFeS is a suitable agent for disrupting biofilm development.

We also evaluated the anti-biofilm efficacy of $Cys_{0.5}$-nFeS using a dental biofilm model on human tooth surface (Fig. 4b). Similar topical treatments with $Cys_{0.5}$-nFeS were employed (at 0, 19, 29, 43, and 53 h) during biofilm formation for 67 h. As shown in Fig. 4c, S. mutans UA159 readily bound and colonized the dental surface, forming a typical biofilm structure with bacterial clusters and the presence of extracellular matrix (see red arrows). However, when treated with $Cys_{0.5}$-nFeS biofilm accumulation and structural organization was markedly disrupted. This result is consistent with processes of bacterial killing and polymeric matrix inhibition. Using a biochemical assays measuring biomass and viability, we found that the biofilms were reduced by 50% in dry weight and 2-log of bacterial viability (CFU biofilm$^{-1}$) in the treated group (Fig. 4d, e).

Since, the formation of biofilm matrix was strongly impaired, we assessed whether the activity of glucosyltransferase B (GtfB) exoenzymes, which is responsible for extracellular glucan matrix synthesis[30,31], was affected. Using established glucan synthesis assays, we found that Cys-nFeS significantly reduced the GtfB activity and polysaccharides synthesis (Fig. 4f). Considering that nFeS is also catalytically active, we examined whether the antibiofilm efficacy could be boosted in the presence of $H_2O_2$. To this end, biofilms were topically treated with $Cys_{0.5}$-nFeS (administered at 19, 29, 43, and 53 h) in combination with 0.5% $H_2O_2$ (administered at 19 and 43 h). We observed a strong reduction of both dry weight and bacterial viability compared to that treated with $Cys_{0.5}$-nFeS or $H_2O_2$ alone (Supplementary Fig. 26). This synergistic effect can be ascribed to three properties, namely releasing polysulfanes from nFeS, enhancing $H_2O_2$ killing

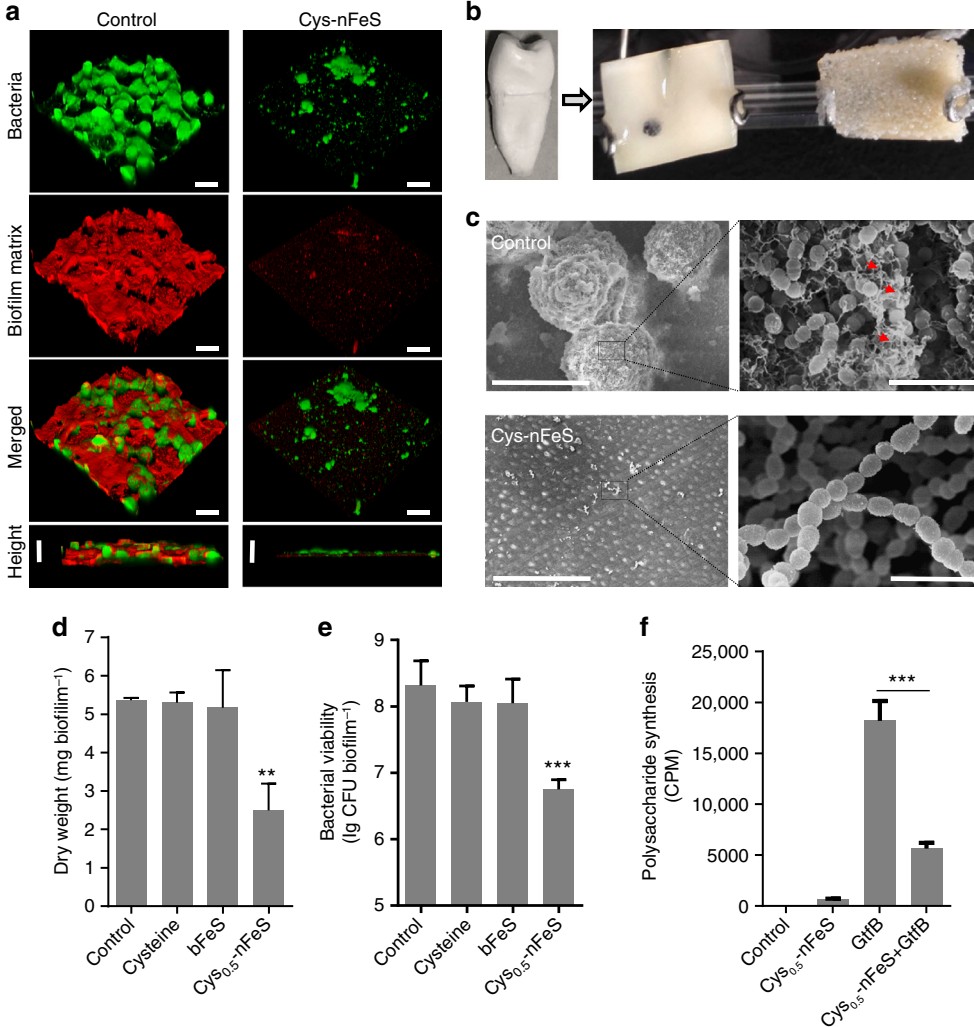

**Fig. 4** Biofilm elimination on tooth surface using nFeS regimen. **a** Confocal 3D image of a *S. mutans* UA159 biofilm treated by Cys-nFeS. Scale bars: 100 μm. **b** *S. mutans* biofilm formed on a dentin surface sectioned from human tooth. **c** SEM image of a *S. mutans* biofilm treated by Cys-nFeS. The red arrows indicate EPS. Left scale bars: 100 μm. Right scale bars: 3 μm. **d, e** Dry weight and cell viability of a *S. mutans* biofilm treated with Cys-nFeS. **f** Inhibition on GtfB activity for polysaccharide synthesis by Cys-nFeS. CPM counts per minute. Data are shown as the mean ± s.d. Statistical significance was assessed using an unpaired Student's two-sided *t*-test compared to the control group. **$p < 0.01$, ***$p < 0.001$ and ****$p < 0.0001$. Mean values and error bars were defined as mean and s.d., respectively. All experiments were performed in triplicate, and representative images are shown

efficacy by enzyme-like activity and the CAR effect as described above. Importantly, Cys-nFeS and the released polysulfanes present in the supernatant showed negligible cytotoxic effects towards human oral keratinocytes (HOK) (Supplementary Fig. 27). Altogether, these results indicate that nFeS is effectively able to disrupt the formation of oral biofilm without any visible toxic side effects.

**Infected-wound healing using topical nFeS treatment**. To further assess its potential as a topical antibacterial alternative, we evaluated the effect of our nFeS on wound healing using an injury-infection model[32]. A model of an infected wound was developed by cutting the back of Balb/c mice to create a wound that was infected with *P. aeruginosa*, a common biofilm-forming pathogen found in chronic wound infections. The nFeS has shown efficient antibacterial activity on *P. aeruginosa* in an in vitro assay (Fig. 2d). SEM characterization showed that the cell morphology of *P. aeruginosa* was changed. In addition, the bacterial flagellum was abrogated, indicating that nFeS damaged the cell integrity of *P. aeruginosa* (Fig. 5a).

Furthermore, the nFeS regimen demonstrated promotion on wound healing in an in vivo model following topical treatment of *P. aeruginosa* infected wound. As shown in Fig. 5b, both $Cys_{0.5}$-nFeS alone or in combination with $H_2O_2$ facilitated wound healing compared to $H_2O_2$ alone. Histological analysis showed that minor or no scabbing occurred in the group treated with $Cys_{0.5}$-nFeS or $Cys_{0.5}$-nFeS + $H_2O_2$ (Fig. 5c). Moreover, the group treated with a combination regimen showed the best recovery from infected wounds, probably because the CAR effect enhances the antibacterial activity of nFeS. Our cell viability assay showed that nFeS (<100 μg mL$^{-1}$) was not toxic to mouse embryonic fibroblasts cells (3T3). In addition, the supernatant from nFeS solution (up to 500 μg mL$^{-1}$) showed no cytotoxicity, indicating that the released polysulfanes possess good biocompatibility (Fig. 5d, e). Importantly, the iron oxide ($Fe_3O_4$) nanoparticles in $Cys_{0.5}$-nFeS regimen showed a long-term favorable effect in stimulating cell proliferation, which may facilitate wound recovery (Fig. 5f). In addition to Cyn-nFeS, DADS-nFeS also exhibited similar effects in the above wound healing tests, indicating all nFeS products have this potency in topical application. (Supplementary Fig. 28). Compared to DADS, DADS-nFeS not

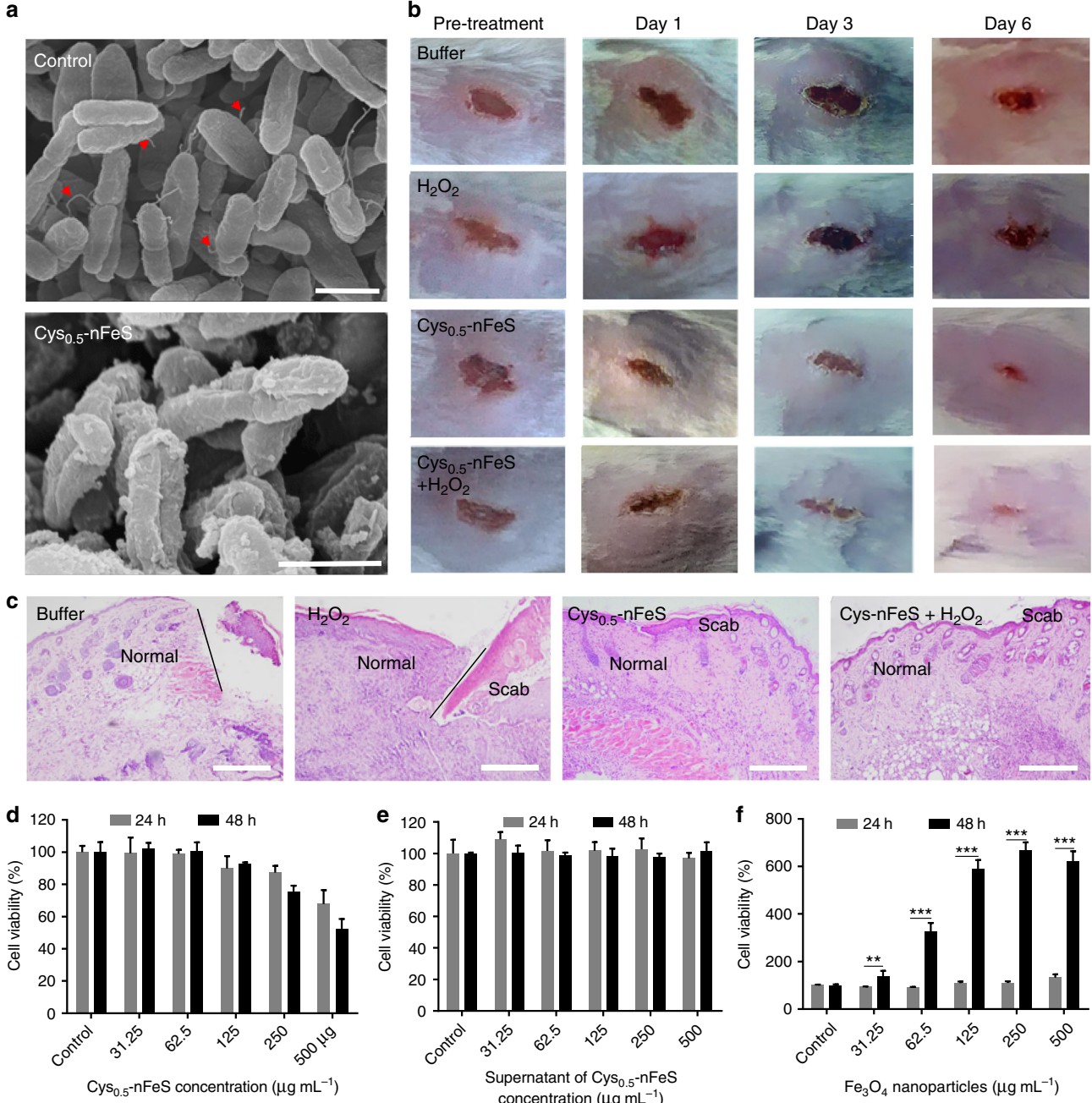

**Fig. 5** Infected-wound healing using nFeS regimen. **a** Morphology of *P. aeruginosa* before (control) and after Cys-nFeS treatment. The red triangles indicate flagella. Scale bars: 1 μm. **b** Photographs of *P. aeruginosa* infected wounds treated with buffer (control), Cys-nFeS, $H_2O_2$, and Cys-nFeS + $H_2O_2$ at different times (five mice in each group). **c** Histologic analyses of the corresponding treated wounds in (**b**) at day 6. Scale bars: 500 μm. **d** Influence of Cys-nFeS on the viability of fibroblast cells. **e** Influence of the supernatant from Cys-nFeS on the viability of fibroblast cells. **f** Stimulation of fibroblast cell proliferation by $Fe_3O_4$ nanoparticles. Data are shown as the mean ± s.d. Statistical significance was assessed using an unpaired Student's two-sided *t*-test compared to the control group. **$p < 0.01$, ***$p < 0.001$ and ****$p < 0.0001$. Mean values and error bars were defined as mean and s.d., respectively. All experiments were performed in triplicate, representative images are shown

only improved wound recovery, but also showed less cytotoxicity (Supplementary Fig. 29). Together, our experimental evaluations clearly demonstrate that nFeS regimen represent an effective therapeutic approach to accelerate wound healing by both reducing bacterial infection as well as stimulating host cell proliferation.

## Discussion

Our work provided a facile strategy to convert natural organosulfur compounds into nFeS with enhanced antibacterial activity against pathogenic and resistant bacteria as well as antibiofilm efficacy. Strikingly, the conversion increased the antibacterial activity several 100-fold compared to the organosulfur precursors. As a potential antibacterial alternative, nFeS demonstrates the following features: (i) nFeS is a highly stable nanomaterial, making it suitable for long-term use. (ii) nFeS displays high dispersibility in aqueous solution, facilitating its reactivity under biological conditions. (iii) nFeS kills bacteria by releasing bactericidal polysulfanes. Such hydrogen polysulfanes have good diffusibility through biological barriers, including cell wall and

biofilm matrix, which could facilitate killing of bacteria residing within a biofilm or invading tissues[33]. (iv) The products of nFeS are odorless, which may merit their popularity in practical use. These features together provide potential advantages for practical applications of nFeS in biomedicine and consumer healthcare.

In our experiments, the conversion efficiency of nFeS showed strong correlation with the type of S-related chemical bond in the organosulfur compounds. The solvothermal condition (200 °C at high pressure) may induce pyrolysis of the S-related bonds to afford free sulfide for nFeS formation. It has previously been shown that thermal degradation occurs in cysteine thus releasing hydrogen sulfide when the temperature is increased to above 200 °C[34]. Thus cysteine and GSH may undergo similar reaction in our solvothermal system. In addition, pyrolysis of the disulfide (DADS, Cystine) or trisulfide (DATS) forms are more favorable as the bond dissociation energies (BDEs) for C–S in these compounds are lower than those in AMS, DAS, and methionine. Therefore, accurate determination (experimentally or computationally) of the BDE of S-related bonds in organosulfur compounds is critical to better understand the conversion of nFeS. Here we chose a computational method to calculate BDE value of C–S and S–S in orgaosulfur compounds. However, the calculated values may differ from those experimentally obtained, because the reaction conditions are considered. For instance, the S–S and allylic C–S bond strengths in DADS were experimentally measured as 62 and 46 kcal mol$^{-1}$, respectively[35]. In comparison, our theoretical calculations resulted in lower values, namely 50.36 and 35.32 kcal mol$^{-1}$, respectively. In addition, S–S bond in trisulfide was measured as 46 kcal mol$^{-1}$ [36], compared to a calculated value of 36.47 kcal mol$^{-1}$ in DATS. These deviations may be ascribed to the different parameters and conditions used in experimental measurements and calculation models. Despite these deviations, the values obtained from both approaches are valuable for understanding the conversion process and mechanism of nFeS formation, and should thereby greatly assist in optimizing nFeS synthesis in solvothermal reaction system.

The antibacterial property of nFeS provides an alternative approach to understanding the bacterial killing mechanism of metal sulfides. Several metal sulfides have been previously reported to efficiently kill bacteria, such as MoS$_2$ and CuS. However, previous research focused primarily on interpreting the antibacterial mechanisms of ROS[32,37–40]. Here we found that the release of polysulfanes plays a critical role in the antibacterial activity. Their release constitutes a coordinated process that involves nanostructure transformation, oxidation and sulfides release. Importantly, our data demonstrate that polysulfanes (H$_2$S$_2$ and H$_2$S$_3$) may be the primary bactericidal molecules to suppress bacterial viability and biofilm formation. We also ruled out the possibility of H$_2$S in antibacterial action, which is an important point since H$_2$S has been reported to enhance drug resistance of bacteria rather than kill bacteria[41]. Together, our results provide insights into the biological functions of hydrogen polysulfanes, as the understanding for these molecules remains limited in biosystem[42,43]. Importantly, such mode of releasing polysulfanes may occur in any materials that contain iron sulfides. Many natural minerals are composed of iron sulfide, such as pyrite, marcasite, pyrrhotite and flint. In particular, flint has been found to be able to release hydrogen disulfanes and hydrogen trisulfanes[44], showing the potential as a donor of polysulfanes that affect microbial ecology. Besides our solvothermal system, nano-iron sulfides can be simply synthesized using a proper sulfur precursor and thus are expected to be able to release polysulfanes[19,45,46]. Therefore, releasing polysulfanes may be a general property for inorganic metal sulfides, which will further expand biomedical applications of sulfur-containing nanomaterials[47].

Future studies will need to investigate the kinetics of polysulfane release, especially when nFeS enters biosystems for in vivo systemic medical applications. Our in vitro antibacterial and antibiofilm tests demonstrated that nFeS not only deformed the bacterial shape when attached on bacterial surface, but also directly inhibited the activity of enzyme excreted by bacteria. These observations suggest that nFeS readily interacts with cell surface and biomolecules at the nano-bio interface. The nanostructure transformation and release of polysulfanes may be affected once biomolecules (proteins, nucleic acids, or metabolites) bind to the surface of nFeS during transit (and thereby prolonged residence in biological fluids) to the targeted site within the body. We have found that protein-rich environment (simulating systemic use) affected the antibacterial effectiveness of nFeS (Supplementary Fig. 30). This effect is likely due to interactions of biomolecules (proteins) on the surface of nFeS thus forming a corona. However, the precise role and characteristics of nano-bio interfaces modulating antibacterial efficacy need to be investigated further, primarily to better understand how the biomolecular corona affects the polysulfanes release within biological fluids[48,49]. These fundamental investigations will be helpful to specify which pathological status is suitable for nFeS treatment.

In summary, our findings offer a nano-conversion strategy to refine and potentiate the performance of natural products for biomedical application. Converting organosulfur compounds into inorganic nano-sulfides significantly improves antibacterial activity and antibiofilm efficacy. This enhanced antibacterial property benefits from nFeS releasing bactericidal polysulfanes. The effective inhibition on *Pseudomonas aeruginosa* and *Staphylococcus aureus* as well as drug-resistant strains, indicates that nFeS could be a potential antibacterial alternative to fight against these pathogens which are in the global priority list released by the WHO[50]. Furthermore, the ability to apply nFeS topically to disrupt pathogenic biofilms and promote wound healing makes it possible for broader applications aimed at preventing or treating biofilm-related infections. Their applications can be extended to disinfect implant devices whose surfaces readily breed bacteria, such as ventilators and blood catheters, often resulting in acute or chronic infections. Therefore, nFeS represents a suitable future option in the fight against pathogenic bacteria with drug resistance or capable of forming intractable biofilms, thus contributing to maintaining high standards of human health presently available.

## Methods

**Materials**. NaCl, NaOAc, glucose, glutaraldehyde were purchased from Sinoparm Chemical reagent (China). FeCl$_3$•6H$_2$O, ethylene glycol, diallyl sulfide (DAS), allyl methyl sulfide (AMS), diallyl disulfide (DADS), diallyl trisulfide (DATS), lysozyme, H$_2$O$_2$ (30%), 2′,7′-dichlorofluorescin diacetate (H$_2$DCFDA), 3,3′,5,5′-tetra-methylbenzidine (TMB), phenylmethyl sulfonyl fluoride (PMSF), and 3-(4,5-dimethylthiazol-2-yl)-2,5-diphenyltetrazolium bromide (MTT) were purchased from Sigma-Aldrich. Tryptone and yeast extract were purchased from Oxoid (UK). Hydroxyapatite discs were purchased from Clarkson Chromatography Products Inc. Alexa Fluor 647-dextran conjugate and SYTO 9 green-fluorescent nucleic acid stain were purchased from Life Technologies. Cysteine, cystine, glutathione (GSH), methionine, ethidium bromide, dimethylsulfoxide (DMSO), and agar were purchased from Sangon Biotech (China). Agarose was purchased from Biowest (Spain). Trans2K Plus II DNA Marker was purchased from TransGen Biotech (China). *Streptococcus mutans* UA159 (ATCC 700610), *Escherichia coli* (*E. coli*, CMCC (B)44102), *Pseudomonas aeruginosa* (*P. aeruginosa*, ATCC 27853), *Staphylococcus aureus* (*S. aureus*, ATCC 29213), *Salmonella enteritidis* (*S. enteritidis*, SC070) and *Candida albicans* (ATCC 10231) were purchased from the Institute of Microbiology of the Chinese Academy of Science. BALB/c mice were obtained from the Vital River Laboratories (Beijing). The human oral keratinocytes (HOK) obtained from ScienCell (SC-2610, ScienCell, USA) and the embryonic mouse fibroblasts (BALB/3T3 clone A31, ATCC CCL-163) obtained from the ATCC were cultured in DMEM medium (Gibco) containing 10% fetal calf serum (Gibco), penicillin (100 U mL$^{-1}$, Sigma-Aldrich) and streptomycin (100 g mL$^{-1}$, Sigma-Aldrich) at 37 °C with 5% CO$_2$.

**nFeS conversion and characterization**. nFeS synthesis was conducted with the typical solvothermal method. Briefly, 0.82 g FeCl$_3$•6H$_2$O was dissolved in 40 mL ethylene glycol. Once the solution was clear, 3.6 g NaOAc and a certain amount of organosulfur compound (AMS, DAS, DADS, DATS, cysteine, cystine, GSH, or methionine) were added with continuous and vigorous stirring for 30 min. The mixture was sonicated for 10 min, transferred to a 50 mL Teflon-lined stainless steel autoclave and reacted at 200 °C for 12 h. After the reaction was completed, the autoclave was cooled to room temperature. The products were washed three times with ethanol and dried at 60 °C for 3 h. The final products were sealed in tube and placed in a desiccator for long-term storage (no >1 month). Morphological and structural characteristics of nFeS were determined with transmission electron microscope (TEM, JEOL JEM-1400 120 kV), scanning electron microscope (SEM, Hitachi S-4800), X-ray diffractometer (XRD, D8 Advance, Bruker AXS, Germany) and vibrating sample magnetometer (VSM, ADE EV7, USA).

**Calculation of C–S bond dissociation energy**. To examine the bond strengths for covalent bonds linking S in the sulfide molecules, we performed density functional theory calculations to obtain the bond dissociation energies ($E_d$). Taking molecule AB as an example, we first fully optimized the structures for molecules AB and its dissociated species (radicals A· and B·) with the B3LYP[51–53]/6-31 G(d,p)[54,55] method. Next, we performed single-point energy calculations with the B3LYP/6-311++G(d,p) method to refine the total energies for the three species. The $E_d$ of bond A–B was calculated via the equation 1.

$$E_d = E_{A-B} - (E_{A·} + E_{B·}),\qquad(1)$$

where $E_{A-B}$, $E_{A·}$, and $E_{B·}$ are the total energies of AB, A·, and B·, respectively. The larger $E_d$, the stronger bond A–B. The solvent effects of water were considered in all calculations using the SMD model[56]. All calculations were performed using the GAUSSIAN 09 program[57].

**Antimicrobial activity of nFeS**. *Streptococcus mutans* UA159 (ATCC 700610) was employed in our experiments. The bacteria were cultured in ultra-filtered (10-kDa cutoff; Millipore, Billerica, MA) tryptone-yeast extract broth (UFTYE, 2.5% tryptone and 1.5% yeast extract, pH 7.0) with 1% glucose at 37 °C and 5% CO$_2$ and collected at the exponential growth phase prior to the experiments. Firstly, *S. mutans* UA159 stored at −80 °C was thawed and innoculabted on UFTYE agar plate for 48 h. Then, monocolony of bacteria was picked into UFTYE liquid medium for 12 h cultivation. The bacterial suspension was diluted 20-fold in fresh UFTYE liquid medium and cultivated at 37 °C and 5% CO$_2$. The growth phase of bacteria was monitored by measuring the optical density (OD) at 600 nm and the value of OD600 reached 1.0 ($10^9$ CFU mL$^{-1}$) within 4 h. Next, 0.1 M sodium acetate (NaOAc, pH 4.5) was used to dilute the bacterial solution to a concentration of $10^6$ CFU mL$^{-1}$ for antibacterial test.

Cys-nFeS stock solution was freshly prepared before experiments. A total of 5 mg of prepared Cys-nFeS powder was first washed three times with 5 mL of distilled water and resuspended into distilled water at the concentration of 5 mg mL$^{-1}$. After autoclaving, the stock solution was diluted at varied concentrations ranging from 0.125 to 1 mg mL$^{-1}$ and added at equal volume to the above bacterial solution for antimicrobial test. A control group with only 0.1 M NaOAc (pH 4.5) was examined at the same time. After incubated for 10 or 30 min, the mixture was diluted by stepwise 10-fold dilution, and 100 μL of the diluted solution was cultured on an agar plate.

Monocolonies of *Escherichia coli* (*E. coli*, CMCC (B)44102), *Pseudomonas aeruginosa* (*P. aeruginosa*, ATCC 27853), *Staphylococcus aureus* (*S. aureus*, ATCC 29213), and its clinically isolated resistant strains *S. aureus* BW15 (anti-Erythromycin, BW indicates that the bacteria was isolated from a burn wound) and *S. aureus* BWMR26 (anti- Ciprofloxacin, Ceftazidime and amikacin, BWMR indicates that the bacteria were isolated from a burn wound with multi-drug resistance (MDR)) on Luria Bertani (LB) solid ager medium were randomly picked and overnight cultured at 37 °C under 180 rpm rotation. The seed solution was diluted 100-fold in fresh liquid LB medium. The bacteria were collected by centrifugation and diluted with NaOAc (0.1 M, pH 4.5) when the OD600 of the bacteria solution reached ~0.8–1.0. The bacteria (100 μL) was mixed with Cys$_{0.5}$-nFeS solutions (100 μL) in 800 μL NaOAc for 30 min. Then, 100 μL of the mixed solution was spread evenly on solid ager medium and cultured at 37 °C for 12–24 h before counting the colony-forming units. *Salmonella enteritidis* (*S. enteritidis*, SC070) was cultured in Sabouraud's medium (1% peptone and 4% glucose), and the antimicrobial experiment was conducted as described above.

**Determination of the internal ROS**. The intracellular ROS level of *S. mutans* stimulated by nFeS was detected by using a 2′,7′-dichlorofluorescin diacetate (DCFH-DA) fluorescent probe. Intracellular DCFH can be oxidized to DCF by ROS. Briefly, after incubating with 10 μM DCFH-DA at 37 °C for 30 min, *S. mutans* UA159 was washed twice with NaOAc, followed by treatment with Cys$_{0.5}$-nFeS. Finally, the ROS level determined as the fluorescence intensity of DCF was measured by a multi-scan spectrum with excitation at 488 nm and emission at 525 nm.

**Determination of internal malondialdehyde**. Intracellular malondialdehyde (MDA) was quantified as an indicator of lipid peroxidation. After treatment with Cys$_{0.5}$-nFeS or 0.1 M NaOAc (control) for 30 min, *S. mutans* UA159 was lysed by lysozyme and proteinase K for MDA measurement. The lysates were centrifuged at $10,000 \times g$ for 10 min to remove the bacteria debris. Supernatants were collected and the levels of lipid peroxidation were determined by reference to a Micro-MDA Assay Reagent Kit (KeyGEN bioTECH, China). For this, 200 μL of thiobarbituric acid (TBA) was mixed with 100 μL of supernatant. The mixture was heated at 95 °C for 40 min. After cooling, the absorbance of the reaction mixture was measured at 532 nm.

**Determination of DNA degradation**. After treatment with Cys$_{0.5}$-nFeS or 0.1 M NaOAc for 30 min, *S. mutans* UA159 was lysed by lysozyme and proteinase K for DNA extraction. Bacterial lysate was collected and processed by using a TaKaRa MiniBEST bacteria Genomic DNA Extraction Kit Ver. 3.0 (TaKaRa, Japan). In the latter experiments, DNA extracts were identified in agarose gel electrophoresis with ethidium bromide staining.

**Bacteria characterization with SEM**. The morphology of *S. mutans* UA159 cells incubated with Cys-nFeS or 0.1 M NaOAc was examined by scanning electron microscope (SEM). First, bacteria suspensions washed with 0.89% (w/v) were resuspended in glutaraldehyde (2%, Sigma-Aldrich) for 4 h at 4 °C under dark conditions. Bacterial cells were then washed and treated with ethanol gradient dehydration (30%, 50%, 70%, 90%, and 100% twice), followed by drying with a critical point dryer and coating with platinum sputter. Finally, the bacterial cells were coated with platinum sputter and analyzed using a scanning electron microscope (Hitachi-S4800). Scanning electron microscope (SEM) images were obtained on a Hitachi S-4800 FE-SEM at a working voltage of 15.0 kV and a working current of 10 μA under magnification of 40K.

**Identification of polysulfanes release from nFeS**. For color observation at different times, 1 mL of nFeS (1.0 mg mL$^{-1}$) in distilled water was incubated in a 24-well plate (Corning Inc., NY). The color of the solution was recorded using a camera. The precipitates from the solution were collected by centrifugation at $3500 \times g$ for 5 min. The collected precipitates were washed with ethanol three times and characterized with SEM for morphological and EDS analysis.

The free sulfide and acid-labile sulfide levels were measured by reversed-phase high-performance liquid chromatography (RP-HPLC) after derivatization with excess monobromobimane (MBB) as stable sulfide-dibimane (SDB) products[20,21,58]. Briefly, nanoparticle materials (1.0 mg mL$^{-1}$, dissolved in PBS) were centrifuged at $3500 \times g$ for 5 min. The supernatants were derivatized using MBB. Thirty microliters of the sample was incubated with 100 μL of Tris-HCl reaction buffer (100 mM Tris, 0.1 mM DTPA, pH 9.5) and 50 μL of monobromobimane (10 mM, dissolved in acetonitrile) under hypoxia (25 °C, 1% O$_2$) for 30 min, and the reaction was stopped by the addition of 50 μL of sulfosalicylic acid (200 mM). The levels of free sulfides were measured by multiple reaction monitoring (MRM) using an Acquity UPLC system coupled to XEVO TQ (Thermo Scientific) with electrospray ionization (ESI(+)). UPLC separation was performed on a BEH C18 column (2.1 mm × 100 mm, 1.7 μm particle size) (Waters, Mississauga, ON, Canada). The mobile phases for LC were water (0.1% TFA, A) and acetonitrile (0.1% TFA, B), and the flow rate was set to 0.3 mL min$^{-1}$. The gradient was as follows: 0 min, 15% B; 0–10 min, 45% B (linear); 10–11 min, 95% B (linear) and hold for 1.0 min; 12–13 min, 15% B (linear) and hold for 2.0 min. Data were collected in MRM mode by screening parent and daughter ions simultaneously, the cone voltage was set depending upon each specific MRM for each metabolite, and the dwell time was automatically set by the MassLynxTM 4.1 software. The optimum collision energy was 30 V. Production ions for MBB derivatives of hydrogen sulfide (H$_2$S), cysteine (cys-SH), disulfanes (HSSH, Cys-SSH), and trisulfanes (H-S-S-SH, Cys-SSSH) levels were 192.1 m/z. Their levels were calculated by the peak area ratio of the signature productions (192.1 m/z) to the corresponding stable S34DB peak (internal standard).

**Enzyme-like activities of nFeS**. The peroxidase-like activity was determined by monitoring the absorbance change at 652 nm on a Microplate Reader (Tecan, Switzerland) in time-course mode at room temperature. The kinetic assays were carried out using 0.2 μg nFeS in 100 μL of reaction buffer (0.1 M NaOAc buffer, pH 4.5) in the presence of H$_2$O$_2$ and TMB. The kinetic analysis of nFeS with H$_2$O$_2$ as the substrate was performed by varying the concentrations of H$_2$O$_2$ with 0.8 mM TMB, and vice versa. The absorbance (652 nm) changes were calculated according to the molar concentration changes of TMB by using a molar absorption coefficient of 39,000 M$^{-1}$ cm$^{-1}$ for TMB-derived oxidation products according to the Beer-Lambert law. All measurements were performed at least in triplicate, and the values were average. The results are given as the mean ± standard deviation (SD). The Michaelis-Menten constant was calculated using Lineweaver–Burk plots of the double reciprocal of the Michaelis-Menten equation (equation 2) by GraphPad Prism (GraphPad Software).

$$v = V\text{max} \times [S]/(K_M + [S]),\qquad(2)$$

where $v$ is the initial velocity, $V$max is the maximal reaction velocity, [S] is the substrate concentration and $K_M$ is the Michaelis–Menten constant.

For the specific activity assay, all the reaction conditions were same, including the sample mass, $H_2O_2$ and TMB concentrations, pH, buffer and temperature. One unit (U) was defined as the amount of enzyme (sample, mg) required to generate 1 μM of TMBox in 1 min at 37 °C in 0.1 M NaOAc (pH 4.5).

The catalase-like activity assay of nFeS was carried out at room temperature by measuring the generated oxygen using a specific oxygen electrode on a Multi-Parameter Analyzer (JPSJ-606L, Leici, China). The generated $O_2$ solubility (unit: mg $L^{-1}$) was measured at different reaction times and the effect of the $H_2O_2$ concentration on the generated $O_2$ was also detected by recording the $O_2$ solubility in NaOAc solution (pH 7.0). The Michaelis–Menten constant was calculated using the same method as mentioned above.

**Biofilm formation**. Biofilms were grown using saliva-coated hydroxyapatite (sHA) discs or dental films[26]. Whole saliva (from one male volunteer, adult) was collected and mixed with equal volume of cold adsorption buffer (50 mM KCl, 1 mM potassium phosphate (0.35 mM $K_2HPO_4$ plus 0.65 mM $KH_2PO_4$), 1 mM $CaCl_2$, 0.1 mM $MgCl_2$. Adjust pH to 6.5). Next, cold 1 mM phenylmethyl sulfonyl fluoride (PMSF) was added to above salvia mixture followed by centrifugation (9000 × $g$, 4 °C, 10 min). Hydroxyapatite discs or dental films were coated with 2.8 mL of the filter-sterilized supernatant of saliva mixture in 24-well plates and vertically suspended in a sterile 24-well plate fixed by a custom-made wire disc holder at 37 °C for 1 h. The bacteria were grown to the mid-exponential phase (optical density at 600 nm was 1.0) as described above. Different concentrations of $Cys_{0.5}$-nFeS were diluted by 0.1 M NaOAc. After washing three times, the saliva-coated sHA discs or dental films were incubated in Cys-nFeS solutions or vehicle (just 0.1 M NaOAc, no Cys-nFeS) for 10 min. Next, sHA discs or dental films were transferred into 2.8 mL of a *S. mutans* UA159 suspension diluted in UFYTE culture medium (pH 7.0) with 1% (w/v) sucrose at a concentration of $10^5$ CFU $mL^{-1}$, and incubated at 37 °C under 5% $CO_2$ for 43 h. The sHA discs or dental films were replenished with fresh culture medium after co-incubation in Cys-nFeS for 10 min. At the end of the biofilm growth period, the biofilms were collected for dry weight measurement, counting of the colony-forming units of *S. mutans* cells cultured on agar plates after dilution to the proper concentration and fluorescence and SEM observation.

**Confocal microscopy of biofilm distribution**. The spatial distribution of biofilm consisting of bacterial cells and extracellular polysaccharide (EPS) matrix was imaged with a confocal scanning microscope. As fluorescently labeled dextran can be incorporated into the EPS-matrix during the course of biofilm formation, a final concentration of 1 μM Alexa Fluor 647-dextran conjugate (647/668 nm; Molecular Probes; 10,000 MW) was added to the UFYTE culture medium (pH 7.0) with 1% (w/v) sucrose throughout the cultivation process, allowing observation of the three-dimensional (3D) structure within intact biofilms. At the end of the biofilm development, the bacterial cells in the biofilm were labeled using 2.5 mM SYTO 9 green-fluorescent nucleic acid stain (485/498 nm; Molecular Probes). Both fluorescent dyes were applied according to standard protocols. The imaging was performed using a Leica TCS SP8 STED confocal microscope (Leica Microsystems, Wetzlar, Germany) with ×20 LPlan N (numerical aperture, 1.05). Three-dimensional scanning software was used to create 3D reconstructions of both the EPS-matrix and bacteria within the biofilm for visualization of the 3D architecture.

**Dentin samples preparation**. Twenty non-carious single-rooted teeth (adult) maintained in phosphate-buffered saline (PBS) solution were used. The crown was sectioned at the level of the cementoenamel junction, and apical portions were ground to obtain root sections. The tooth specimens were then vertically sectioned along the mid-sagittal plane into 2 halves (mesial and distal). The root canal lumen was flattened using progressive 1000–4000 grit silicon carbide papers. Silicon carbide paper was used on the external root surface to ensure that the surfaces were parallel to the root canal wall in the section. The samples were then ultrasonically cleaned in deionized water for 30 min to obtain root canal dentin specimens without the presence of the smear layer. The samples were stored in double-distilled water until further use. The study was approved by the Sichuan University Ethics Board (WCHSIRB-D-2016-140).

**GtfB activity in the present of nFeS**. The influences of the presence of Cys-nFeS on the synthesis of glucans by GtfB was determined in solution phase[59–61]. The GtfB enzyme was prepared and purified using hydroxyapatite column chromatography[62]. For glucan synthesis in the solution phase, GtfB (10 units) was mixed with $Cys_{0.5}$-nFeS (0.5 mg $mL^{-1}$), and incubated with ([14 C]glucosyl)-sucrose substrate (0.2 μCi $mL^{-1}$; 200 mM sucrose, 40 μM dextran 9000, 0.02% sodium azide in adsorption buffer, pH 6.5) to reach a final concentration of 100 mM sucrose (reaction volume of 200 μL), and incubated at 37 °C. After 4 h incubation, the total amount of glucans formed was measured by scintillation counting. The control contained the same reaction mixture with adsorption buffer but without Cys-nFeS. The solutions were incubated at 37 °C with rocking for 4 h to allow glucan synthesis. Subsequently, the glucans were precipitated with ice-cold ethanol (final concentration of 70%) for 18 h at 4 °C. The radiolabeled glucans were then determined by a scintillation counting.

**Cell viability assay of human oral keratinocytes (HOK)**. HOK cells (SC-2610, ScienCell, USA) were seeded into a sterile 96-well plate (Corning Inc., NY) at $10^4$ well$^{-1}$ in the medium (DMEM with 10% FBS) and followed to attach for 24 h at 37 °C under 5% $CO_2$. The cells were treated with a series of $Cys_{0.5}$-nFeS solutions (31.25, 62.5, 125, 250, and 500 μg $mL^{-1}$) for 10 min, and then, $Cys_{0.5}$-nFeS was washed away. Fresh medium was added, and the cells were cultured for 24 and 48 h. In addition, the cells were also treated with supernatant from the corresponding $Cys_{0.5}$-nFeS solution at the same concentrations for 24 and 48 h. Finally, the cells were added with a final concentration of 0.5 mg $mL^{-1}$ MTT and incubated for 4 h. The medium was removed, and 150 μL of DMSO was added. The absorbance was determined at 490 nm, and the cell viabilities were expressed as a percentage of the control values.

**Mouse injury model**. All animal studies were performed following a protocol approved by the Institutional Animal Care and Use Committee of the Institute of Biophysics, Chinese Academy of Sciences. Twenty-five 6–8-week-old-male Balb/c mice were purchased from Vital River. The back of each mouse was cut and injected with $10^7$ CFU of *P. aeruginosa* bacteria to build the infected wound model according to a study reported by Cao et al.[32]. The mice were then divided into five groups (five mice per group). The mice with infected wounds in the different groups were treated with the gauze containing 10 μL of NaOAc buffer, 100 μM $H_2O_2$, 100 μg $mL^{-1}$ of $Cys_{0.5}$-nFeS, or 100 μM $H_2O_2$ + 100 μg $mL^{-1}$ of $Cys_{0.5}$-nFeS. The wounds were photographed, and bandages were changed at 24 h intervals. After 6 days of therapy, the mice were killed, and the related wound tissues were collected and fixed by formalin. The wound tissues were then paraffined, sectioned, and analyzed via Hematoxylin-Eosin (HE) staining. The tissue sections were examined by a Nikon Eclipse Ci microscope in bright-field mode.

**Cell viability assay of fibroblast cells**. Embryonic mouse fibroblasts (BALB/3T3 clone A31, ATCC CCL-163) were seeded into sterile 96-well plates at $10^4$ well$^{-1}$ in the medium (DMEM with 10% FBS) and followed to attach for 24 h at 37 °C under 5% $CO_2$. The cells were treated with a series of the supernatant and precipitate of $Cys_{0.5}$-nFeS solutions (31.25, 62.5, 125, 250, and 500 μg $mL^{-1}$) for 24 or 48 h. Finally, cells were added at a final concentration of 0.5 mg $mL^{-1}$ MTT and incubated for 4 h. The medium was removed, and 150 μL of DMSO was added. The absorbance was determined at 490 nm, and the cell viabilities were expressed as a percentage of the control.

**Statistic methods**. The significance of the data in Figs. 2a–j, 3e, 4d–f, and 5f was analyzed according to unpaired Student's two-sided $t$-test. **$p < 0.01$, ***$p < 0.001$, and ****$p < 0.0001$. The enzymatic kinetics was analyzed with Michaelis–Menten equation by Graphpad Prism 7.0. The samples/animals were randomly allocated to experimental groups and processed for blind evaluation.

## Data availability

All data are available from the authors upon reasonable request.

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

## Acknowledgements

We thank the Testing and Analysis Centre at Yangzhou University for the nanomaterial characterizations. This work was supported by the National Natural Science Foundation of China (Grant No. 81671810), the Jiangsu Provincial Basic Research Program for the Natural Science Foundation (Grant No. BK20161333), the Foundation of the Thousand Talents Plan for Young Professionals and Jiangsu Specially-Appointed Professor, and in part by the National Institutes of Health/National Institute of Dental and Craniofacial Research (NIH/NIDCR) R01 DE025848.

## Author contributions

L.G. conceived and designed the experiments. Q.L., Z.Q., D.L., Y.T. and J.Q. synthesized and characterized the materials. Z.X., Z.Q, D.L. and X.S. characterized sulfide release and

identification. Q.L., Z.Q. and Y.G. contributed to the enzyme kinetics assay. Z.X. and Z.Q. performed the bacteria and biofilm assays and obtained the images. J.H. and J.X. provided the dentin sample. X.G. calculated the energy of the C–S bond in organosulfur compounds. Y.L. and H.K. contributed to the dental biofilm assay. Y.H., Z.X. and J.J. analyzed cell toxicity and K.F. performed wound healing experiment. L.G. and Z.X. analyzed the data and wrote the manuscript. H.K. and X.Y. helped revise the manuscript. All authors discussed the results and commented on the manuscript.

## Additional information

**Competing interests:** The authors declare no competing interests.

