## [Peer Review File · Nature Communications]

Reviewers' comments:

Reviewer #1 (Remarks to the Author):

This manuscript deals with converting natural organosulfur into inorganic polysulfide as a powerful strategy to combat bacterial infections. The authors developed a strategy to synthesize nFeS nanosheets and provided comprehensive evaluations on their characteristics and antibacterial properties to prove their claims. Although the outcomes are interesting, a major revision is required before publication of this paper:

- The role and characteristics of nano-bio interfaces on the observed antibacterial efficacy of the nanosheets are ignored. The others should carefully consider the role of biomolecular corona and its effects on the physicochemical properties on the platform which directs their interactions with biosystems.
- The produced nanosheets were washed several times with ethanol only. This reviewer wonders why water has not been used in the washing procedure? Stability of the products in water should not be a problem here, as the nanosheets were suggested to be highly stable in water (as claimed in the paper); therefore, the reason of using ethanol is not justified in the text and is highly questionable.
- According to the authors' observations, active components in the supernatant of nFeS played the critical antibacterial role. Therefore, to have better/mechanistic understanding of the role of individual compounds, the authors should monitor the release of other compounds (e.g., hydrogen polysulfides) from nFeS and consider their antibacterial properties.
- Some sections of the paper are vaguely presented and thus can be confusing to the readers. For example, on page 7, line 137, the authors claimed that "The product from Cys1.0 only contained nanosheets and hexagonal structures whereas Fe₃O₄ nanoparticles were present in the product from Cys0.5". However, the authors used Cys0.5-nFeS in the majority of the experiments (although it seems to this reviewer that they should have used pure Cys1.0-nFeS instead).
- Previous seminal reports in the field (e.g., synthesis of FeS nanosheets and antibacterial nanotechnologies) are not acknowledged in this paper. The references should be carefully updated in the revised manuscript; in addition, pros and cons of the proposed approach compared to the reported ones should be discussed.
- Critical information on the biosystems and nanobio interfaces are missed in the paper; this will cause substantial issues in reproducing the results by other researchers. The required information should be carefully added to the revised manuscript (the following articles might be helpful, as they comprehensively covered the essential information which should be included in reports: Trends Biotechnol. 2018, DOI: 10.1016/j.tibtech.2018.02.014; Nat. Rev. Mater. 2016; DOI: 10.1038/natrevmats.2016.14).

Reviewer #2 (Remarks to the Author):

This manuscript presents some very nice, new organosulfur chemistry and therefore merits publication in Nature Communications if several corrections can be made.

First, the calculated C–S bond dissociation energies in Figure 1 do not agree with experimental values, which should be more reliable. For example, see J. Am. Chem. Soc. 1988, 110, 7813-7827 for experimental values for the S–S and allylic C–S bond strengths in diallyl disulfide of 62 and 46 kcal mol⁻¹, respectively. The S–SS bond is reported as 46 kcal mol⁻¹ [Pickering, T. L.; Saunders, K. L.; Tobolsky, A. V. In The Chemistry of Sulfides; Tobolsky, A. V., Ed.; Interscience: New York, 1968; p 61.] The authors should search the literature for experimental values for dissociation

energies to confirm or correct computed values, which may not be correct, particularly given the disagreement in the above experimental and computed values. The authors' computational method may not be reliable!

Second, the release of H₂Sn from iron sulfides in the form of flint minerals has already been reported. This prior work must be fully discussed by the authors: see: DOI: 10.1021/acs.jafc.6b03938

J. Agric. Food Chem. 2016, 64, 9033–9040. On line 174, 376 and elsewhere the term “polysulfide” is incorrectly used. Compounds of type H₂Sn should be termed hydrogen trisulfane and hydrogen polysulfanes, as used in the JAFc paper, rather than “hydrogen polysulfides”. However anionic Sn can still be called polysulfide.

Third, more comprehensive recent review references could be cited regarding Allium compounds, for example, Eric Block, Garlic and Other Alliums: The Lore and the Science, Royal Society of Chemistry, Cambridge, UK: 2010 and Chemistry Industry Press, Beijing, China: 2017 (Chinese Edition).

Fourth, the reference style should be made uniform, following the journal style, with regard to capitalizing first letters of words in titles and names of publications – either capitalize only the first letter of the first word or capitalize all first letters of words. There also seem to be various stylistic errors in references 24, 26, 31, 33 and 34 (subscripting; shouldn't names of organisms in paper titles be italicized, as on line 149?). In the caption to Figure 3, MBB, TCEP and CAR should be defined even if defined elsewhere; line 87, NaOAc is the correct abbreviation for sodium acetate.

Finally, the language in this manuscript needs to be carefully corrected in a revision. For example, the title of the paper reads poorly – there is no such thing as “natural organosulfur” Do the authors mean “natural organosulfur compounds?” Similarly, on line 29 the term beginning the abstract, “Natural substance” is incorrect as used; “The use of natural substances” would make more sense. “Natural organosulfurs” (line 31, 360) and later “organosulfur (lines 40, 43, 48, 144, 148, 410) make no sense – “natural organosulfur compounds” and “organosulfur compounds” is preferable. Throughout the paper “natural organosulfur” incorrectly appears; line 38 “teeth” not “tooth”; line 54 “Welsh” is a proper name; errors/typos lines 70 (“perform high antibacterial potency” doesn't make any sense); 71, “organosulfur compounds”; 79 “wound”; throughout page 4: “solvothermal method” doesn't make any sense and needs to be defined for the general reader; 106 “diallyl sulfide” is two words; 202, “devirratization” doesn't make sense; line 279 “which may be benefited from the high permeability of the released polysulfides” doesn't make sense; line 335, the term “slashing” impacts cruelty to animals – “cutting” would be better; line 362 “organosulfur compounds form”; lines 372-374 are poorly written and difficult to understand -- “large” not “big” and other changes needed.

Reviewer #3 (Remarks to the Author):

In the manuscript the authors present an interesting approach of converting natural organosulfur into inorganic polysulfide with antimicrobial potency. They provide the antimicrobial activity of iron sulfides obtained from various organosulfur sources. But for any further investigation their choice of a standard molecule is cysteine. The work would be much more convincing and of interest to others in the community if the whole study is done with nature-derived compounds, such DATS or DADS as the initial assay was done.

Therefore the current study needs additional experiments to be performed with paying more attention to the details such proper controls (known natural antimicrobial agent with known activity), always providing concentrations (Fig. 2a) and statistical significance (is missing in all experiments).

The style of writing has to be improved with the clear message given in the conclusion.

We really appreciated the positive comments from the reviewers. All the comments and suggestions were critical and constructive, which helped improve the quality of our work and manuscript. We have made the corresponding revisions and updated new data in the revised manuscript and supplementary information (marked in red color). We adjusted the new title of manuscript as “Converting organosulfur compounds to inorganic polysulfides against resistant bacterial infections”. Below please find our replies to all questions posed by the reviewers.

Reviewer #1

This manuscript deals with converting natural organosulfur into inorganic polysulfide as a powerful strategy to combat bacterial infections. The authors developed a strategy to synthesize nFeS nanosheets and provided comprehensive evaluations on their characteristics and antibacterial properties to prove their claims. Although the outcomes are interesting, a major revision is required before publication of this paper:

Response: We appreciated the positive comment on our work.

1. The role and characteristics of nano-bio interfaces on the observed antibacterial efficacy of the nanosheets are ignored. The others should carefully consider the role of biomolecular corona and its effects on the physicochemical properties on the platform which directs their interactions with biosystems.

Response: We appreciate the reviewer pointing out this important question. The interaction occurring at the interface of nFeS and biomolecules is relevant for nFeS bioactivity in biological system. Our antibacterial experiments were topically conducted with saliva-coated surface for biofilm elimination and *in vivo* skin for wound healing, which takes into consideration the nano-biointerfaces. We assess nFeS with brief topical exposure where the effects are on site. So the protein corona may cause lesser influence compared to systemic administration (when the resident time within biological fluids is very high). But we recognized that we failed to provide experimental details and discuss the importance of nano-biointerfaces and related issues. We have now included discussion about nano-bio interfaces and performed additional experiments to investigate the potential effects of biomolecular corona on antibacterial efficacy, and added related information in the revised manuscript.

Furthermore, we also assessed antibacterial activity of nFeS incubated in a protein-rich medium (for cases simulating systemic use vs topical as proposed here). We observed reduction of antibacterial activity when incubated in mammalian cell culture medium (RPMI-1640) plus 10% FBS and in the plasma of mouse serum (Supplementary Figure 30), yet it was still able to achieve >1 log viability reduction. This observation is likely due to interactions of biomolecules (proteins) on the surface of nFeS to form corona as

pointed out by the reviewer. The presence of protein corona may not only affect the release of polysulfanes from nFeS, but also reduce the bioavailability of polysulfane as the thiol groups in protein may directly react with polysulfane. Our future study will focus on investigating the potential interaction and influence of biosystem on nFeS using *in vitro* serum or cell system and *in vivo* animals.

We have added these points in the revised manuscript while emphasizing that additional studies are needed for this complex and relevant point, especially for systemic use (in page 21 of revised manuscript).

2. The produced nanosheets were washed several times with ethanol only. This reviewer wonders why water has not been used in the washing procedure? Stability of the products in water should not be a problem here, as the nanosheets were suggested to be highly stable in water (as claimed in the paper); therefore, the reason of using ethanol is not justified in the text and is highly questionable.

Response: We thank the review pointing out this question. We made the nFeS products by solvothermal reaction which using ethylene glycol as the solvent. When the reaction was completed in the sealed Teflon reactor, we collected the products and discarded the supernatant of ethylene glycol. Ethanol washing is a routine step to remove the remaining ethylene glycol and other chemicals on the surface of nanomaterials. Ethanol allows nFeS to disperse completely and volatilizes very quickly during drying step, which is also good for sample preparation in SEM and TEM tests. Ethanol washing was used only in above step.

In the antibacterial test, we always used distilled water to wash nFeS product 3 times and then made a stock of nFeS in water. The nFeS stock was further sterilized in autoclave and diluted for antibacterial test. Therefore, nFeS is feasible to be treated and dissolved in aqueous solution with high stability.

We appreciate that the reviewer pointing out this question and have clarified it in the experimental section (marked in red text). Please see “Antimicrobial activity of nFeS” in the revised method.

3. According to the authors' observations, active components in the supernatant of nFeS played the critical antibacterial role. Therefore, to have better/mechanistic understanding of the role of individual compounds, the authors should monitor the release of other compounds (e.g., hydrogen polysulfides) from nFeS and consider their antibacterial properties.

Response: Thanks for the reviewer's suggestion. In our paper we proposed that the antibacterial ability of nFeS is releasing active hydrogen polysulfanes during the nanostructure transformation from nanosheets to nanoparticles. We used HPLC-MS to analyze the ingredients in the supernatant and confirmed there were hydrogen sulfide, hydrogen persulfide and hydrogen trisulfide with minor cysteine-derived sulfide (Cys-SSH and Cys-SSSH) (Fig. 3c, 3d and supplementary Fig. 14 and Supplementary Table 2). We excluded hydrogen sulfide using NaHS as a donor which did not show antibacterial activity (Please see Fig 2b). We further confirmed that the supernatant lost bactericidal ability after the treatment with TCEP which specifically break S-S bond (Fig. 3e), indicating polysulfanes contributes to the antibacterial reactivity.

HPLC-MS results showed that the release of polysulfanes was time-dependent and hydrogen disulfane and hydrogen trisulfane were dominant in the supernatant (Supplementary Table 2). We compared the antibacterial efficiency between nFeS and its supernatant. The results showed that the supernatant also exerted considerable antibacterial activity similar as the nFeS solution.

Therefore, the above data demonstrated the mechanism of nFeS for antibacterial activity. For future study we will explore the strategy to quantify the amount of hydrogen polysulfanes in the supernatant, prepare pure product of hydrogen disulfane (H_2S_2) and hydrogen trisulfane (H_2S_3) and provide direct evidence of antibacterial activity of hydrogen polysulfanes using *in vivo* models.

4. Some sections of the paper are vaguely presented and thus can be confusing to the readers. For example, on page 7, line 137, the authors claimed that "The product from Cys_{1.0} only contained nanosheets and hexagonal structures whereas Fe₃O₄ nanoparticles were present in the product from Cys_{0.5}". However, the authors used Cys_{0.5}-nFeS in the majority of the experiments (although it seems to this reviewer that they should have used pure Cys_{1.0}-nFeS instead).

Response: Thanks for pointing this out. Among cysteine-derived nFeS, we found that Cys_{0.5}-nFeS exhibited the best antibacterial activity and enzyme-like activity. Then we investigated the performance of Cys_{0.5}-nFeS in antibacterial action, polysulfane release, enzyme-like catalysis and topical applications in biofilm elimination and wound healing. In addition, Cys_{0.5}-nFeS showed paramagnetic property which makes it easy to use in the experiment. The presence of Fe₃O₄ nanoparticles in the product of Cys_{0.5}-nFeS showed positive stimulation on fibroblast cell proliferation, which may be beneficial for wound healing (Fig. 5f).

In contrast, DADS_{1.0}-nFeS performed the best activities among DADS-derived nFeS. To confirm the antibacterial equivalence, we did the same tests and found DADS_{1.0}-nFeS performed very similar antibacterial activity and enzyme-like activity and exhibited

comparative efficacy in biofilm elimination and wound healing (please see Supplementary Figure 10, Supplementary Figure 21, Supplementary Figure 24 and Supplementary Figure 28). We have clarified these points in the revised manuscript.

5. Previous seminal reports in the field (e.g., synthesis of FeS nanosheets and antibacterial nanotechnologies) are not acknowledged in this paper. The references should be carefully updated in the revised manuscript; in addition, pros and cons of the proposed approach compared to the reported ones should be discussed.

Response: Thanks for the reviewer's suggestions. We have added references to acknowledge the previous reports in the field and discussed them in the revised main text (marked in red).

6. Critical information on the biosystems and nanobio interfaces are missed in the paper; this will cause substantial issues in reproducing the results by other researchers. The required information should be carefully added to the revised manuscript (the following articles might be helpful, as they comprehensively covered the essential information which should be included in reports: Trends Biotechnol. 2018, DIO: 10.1016/j.tibtech.2018.02.014; Nat. Rev. Mater. 2016; DOI: 10.1038/natrevmats.2016.14).

Response: We thank the reviewer's suggestion and information. Also please check our response #1. We agree that it is an important and relevant point to consider when assessing antibacterial behavior of nFeS in biosystem, especially for systemic use. To give accurate information, we provided a section of "Materials" to introduce all the source and number of chemicals, bacteria, cells and animals used in our experiment (please see the revised method). Other biological materials, such as saliva-coated hydroxyapatite surface and dentin slide to mimic oral environment for biofilm tests were also included. Furthermore, the references were also included.

We appreciate that the reviewer pointed out the importance of nano-bio interfaces, and inclusion of information for all materials and experimental conditions, which will be helpful for the reproducibility of the results by other scientists and the translation to clinical or industrial application.

Reviewer #2:

This manuscript presents some very nice, new organosulfur chemistry and therefore merits publication in Nature Communications if several corrections can be made.

Response: We appreciate the positive comments from the reviewer.

1. First, the calculated C–S bond dissociation energies in Figure 1 do not agree with experimental values, which should be more reliable. For example, see J. Am. Chem. Soc. 1988, 110, 7813-7827 for experimental values for the S–S and allylic C–S bond strengths in diallyl disulfide of 62 and 46 kcal mol⁻¹, respectively. The S–SS bond is reported as 46 kcal mol⁻¹ [Pickering, T. L.; Saunders, K. L.; Tobolsky, A. V. In The Chemistry of Sulfides; Tobolsky, A. V., Ed.; Interscience: New York, 1968; p 61.] The authors should search the literature for experimental values for dissociation energies to confirm or correct computed values, which may not be correct, particularly given the disagreement in the above experimental and computed values. The authors' computational method may not be reliable!

Response: We thank the reviewer for pointing out the inconsistency between the experimental and computational bond dissociation energies (BDE). We also appreciate that the reviewer provided the information about experimental data of C-S and S-S bonds, which is very helpful to understand conversion of organosulfur compounds in our solvothermal system. We have recalculated the BDE using a revised approach. All BDE data have thus been updated in Fig. 1. The computational method section has also been accordingly updated.

We realize that the absolute values of the newly calculated BDE still have a systematical difference of about 10 kcal mol⁻¹ from those of the experimental values. However, there are several similarities in the relative differences for both experimental measurement and computational calculation. First, the experimental values of BDE for S–S and allylic C–S in DADS are 62 and 46 kcal mol⁻¹, respectively, indicating C-S bond is weaker than S-S bond by 16 kcal mol⁻¹. In comparison, the calculated values of BDE for S-S and C-S are 50.36 and 35.32 kcal mol⁻¹, respectively, giving a 15.04 kcal mol⁻¹ difference between S-S and C-S. Therefore, the difference value of BDE between S-S and C-S from calculation is relatively close to that from experimental measurement.

Second, the experimental S–SS bond in DATS was reported as 46 kcal mol⁻¹, giving 16 kcal mol⁻¹ weakness than that for S-S bond measured in DADS. Our calculation gives the value to be 36.47 kcal mol⁻¹ for S-SS bond, showing 13.89 kcal mol⁻¹ weakness than that for S-S bond calculated in DADS. Although absolute values are lower than those from experimental measurement, calculation result shows the similar relative weakness for S-SS bond in DATS compared to S-S bond in DADS.

Third, the value of 46 kcal mol⁻¹ for S-SS in DATS is same as that for C-S in DADS according experimental measurement. On the basis of calculation, the value of S-SS bond in DATS is 36.47 kcal mol⁻¹, which is also close to the values of 38.68 kcal mol⁻¹ for C-S in DATS and 35.35 kcal mol⁻¹ for C-S in DADS.

Therefore, C-S bond is easier to break than S-S in DADS, and S-SS bond in DATS becomes weaker than S-S in DADS, which both are supported by experimental measurements and computational calculations. Although the absolute values of BDE for S-related bonds in DADS and DATS are different, the variation tendency from the computational values are in good agreement with that from the experimental ones.

We think the calculation partially provides basic information to understand why organosulfur compounds containing S-S and S-S-S have higher conversion efficiency than those with C-S-C in our present work. We will keep working on determining the BDE values by combining computational calculation and experimental measurement, which may provide guidance for nFeS conversion from different organosulfur sources in the future.

2. Second, the release of H_2Sn from iron sulfides in the form of flint minerals has already been reported. This prior work must be fully discussed by the authors: see: DOI: 10.1021/acs.jafc.6b03938, J. Agric. Food Chem. 2016, 64, 9033–9040. On line 174, 376 and elsewhere the term “polysulfide” is incorrectly used. Compounds of type H_2Sn should be termed hydrogen trisulfane and hydrogen polysulfides, as used in the JAFC paper, rather than “hydrogen polysulfides”. However anionic Sn can still be called polysulfide.

Response: Thanks for the important information from the reviewer. We have discussed the potential antibacterial activity of flint. We adjusted the name for polysulfanes, especially for hydrogen disulfane and hydrogen trisulfane. Please see the revised manuscript.

3. Third, more comprehensive recent review references could be cited regarding Allium compounds, for example, Eric Block, Garlic and Other Alliums: The Lore and the Science, Royal Society of Chemistry, Cambridge, UK: 2010 and Chemistry Industry Press, Beijing, China: 2017 (Chinese Edition).

Response: Thanks for the important information. We carefully read the book to learn the history and research on Garlic. We have added the reference to the revised manuscript.

4. Fourth, the reference style should be made uniform, following the journal style, with regard to capitalizing first letters of words in titles and names of publications – either capitalize only the first letter of the first word or capitalize all first letters of words. There

also seem to be various stylistic errors in references 24, 26, 31, 33 and 34 (subscripting; shouldn't names of organisms in paper titles be italicized, as on line 149?).

Response: We have revised the uniform style in the revised manuscript.

In the caption to Figure 3, MBB, TCEP and CAR should be defined even if defined elsewhere; line 87, NaOAc is the correct abbreviation for sodium acetate.

Response: We have added the information and corrected it in the revised manuscript.

5. Finally, the language in this manuscript needs to be carefully corrected in a revision. For example, the title of the paper reads poorly – there is no such thing as “natural organosulfur”. Do the authors mean “natural organosulfur compounds?” Similarly, on line 29 the term beginning the abstract, “Natural substance” is incorrect as used; “The use of natural substances” would make more sense. “Natural organosulfurs” (line 31, 360) and later “organosulfur (lines 40, 43, 48, 144, 148, 410) make no sense – “natural organosulfur compounds” and “organosulfur compounds” is preferable. Throughout the paper “natural organosulfur” incorrectly appears;

Response: Thanks for the reviewer's suggestions. We have made the correction and used organosulfur compounds in the whole text.

line 38 “teeth” not “tooth”; line 54 “Welsh” is a proper name; errors/typos lines 70 (“perform high antibacterial potency” doesn't make any sense); 71, “organosulfur compounds”; 79 “wound”; throughout page 4: “solvothermal method” doesn't make any sense and needs to be defined for the general reader; 106 “diallyl sulfide” is two words; 202, “deviratization” doesn't make sense; line 279 “which may be benefited from the high permeability of the released polysulfides” doesn't make sense; line 335, the term “slashing” impacts cruelty to animals – “cutting” would be better; line 362 “organosulfur compounds form”; lines 372-374 are poorly written and difficult to understand – “large” not “big” and other changes needed.

Response: We appreciate the reviewer pointing out the issues. We have corrected and revised them in the revised manuscript.

Reviewer #3:

1. In the manuscript the authors present an interesting approach of converting natural organosulfur into inorganic polysulfide with antimicrobial potency. They provide the antimicrobial activity of iron sulfides obtained from various organosulfur sources. But for any further investigation their choice of a standard molecule is cysteine. The work would be much more convincing and of interest to others in the community if the whole study is done with nature-derived compounds, such DATS or DADS as the initial assay was done. Therefore the current study needs additional experiments to be performed with paying more attention to the details such proper controls (known natural antimicrobial agent with known activity), always providing concentrations (Fig. 2a) and statistical significance (is missing in all experiments).

Response: We appreciate the positive comment on our work from the reviewer. We carefully investigated the antibacterial activity of DADS-nFeS, and additional data was provided in the revised supplementary information. We also make the modification to mark concentrations in the legend of Fig. 2a and provided statistical analysis for all data in the revised manuscript (marked in red).

In the revised manuscript we carefully investigated the antibacterial activity of DADS-nFeS and compared the difference between DADS-nFeS and DADS. As expected, DADS-nFeS showed similar antibacterial activity as Cys-nFeS for *E.coli*, *S. mutans*, *P. aeruginosa* and *S. aureus* as well as the resistant strains (Supplementary Figure 10 a-f), whereas DADS showed negligible antibacterial activity to these pathogens (Supplementary Figure 11). DADS-nFeS induced high ROS, high MDA and DNA degradation in bacterial killing process (Supplementary Figure 10 g-i). Dental biofilm test proved that DADS-nFeS has high effect on biofilm elimination (Supplementary Figure 24) and animal test showed that DADS-nFeS also promoted wound healing (Supplementary Figure 28). In addition, similar color change and nanostructure transformation were observed in DADS-nFeS solution (Supplementary Figure 12), indicating polysulfanes release from DADS-nFeS. DADS-nFeS also performs high enzyme-like activity which can synergize the antibacterial activity (Supplementary Figure 21). These results demonstrated that DADS-nFeS have similar physicochemical property and antibacterial activity as those in Cys-nFeS.

However, we found that DADS exhibited poor solubility in ethylene glycol and water (Supplementary Figure 4). The poor solubility impacted DADS-nFeS synthesis in solvothermal reaction as it was hard to accurately evaluate the correlation between DADS amount and DADS-nFeS formation. Due to the same reason, DADS showed very limited antibacterial activity tested in aqueous solution. In addition, our new data found that DADS had strong cytotoxicity to mammalian cells (3T3) (Supplementary Figure 29).

Therefore, although DADS-nFeS showed equivalent potency as Cys-nFeS, we think that it is better to use cysteine as standard molecule for our study. Several reasons are considered in our experiments. First and importantly, cysteine is a common organosulfur compound in nature. It is not only one of the most abundant amino acids in biosystem, but also the precursor for many other organosulfur compounds and metabolites. For

instance, in garlic, alliin, derived from cysteine, is converted into allicin which further derive DADS, DATS, DAS and AMS. In addition, methionine, cysteine and GSH are also derived from cysteine. Therefore, cysteine can be considered a source of all these organosulfur compounds tested in our experiments. Second, compared to DADS or DATS, cysteine is odorless, stable and water-soluble, which is more suitable as a model molecule to investigate the correlation between sulfur amount and nFeS. Altogether, cysteine appears to be a more feasible standard molecule for the investigation. We have included this point in the revised manuscript.

2. The style of writing has to be improved with the clear message given in the conclusion.

Response: Thanks for the reviewer's suggestion. We have revised the discussion and conclusion in the revised manuscript.

REVIEWERS' COMMENTS:

Reviewer #1 (Remarks to the Author):

The authors carefully considered the comments and performed the requested additional experiments in the revised version of the manuscript. I would publish this paper.

Reviewer #2 (Remarks to the Author):

While the authors have adequately addressed the technical concerns, the manuscript requires careful editing to improve the language and clarity and correct numerous grammatical errors. The authors should use the services of an editor expert in English to make the manuscript acceptable for publication. A few, but not all of the problems are noted below.

Line 67 (while "a" small number ...)

Lines 74, 75

Line 88 "sulfur element"??? do you mean "elemental sulfur"

Line 98 -- do not capitalize "selected-area electron diffraction"

Lines 114-116 "kcal"; "formed" doesn't make sense; meaning of pairs of numbers "(69.358/50.69) and (66.90/70.69) is unclear -- why pairs of numbers??

Lines 123-130 -- many mistakes of grammar, etc. Thorough editing needed here.

Line 159 "crystalline"

Line 187-188 poor English -- rewrite

Line 224 doesn't make sense

Line 258 Poor English -- rewrite ("Regarding to the components")

Line 429 doesn't make sense ("Inconsistence")

Line 441 Poor English "But..."

Line 451 Poor English ("releasing mode of polysulfanes")

Line 462 Poor English "destructured"

Main References and Supplementary References: proper journal style should be followed throughout using abbreviations and capitalized first letters for journal names,

Reviewer #3 (Remarks to the Author):

The authors have significantly improved the quality of written manuscript by providing all necessary and previously missing references, paying more attention to the details such providing concentrations and statistical analyses. They also included additional and requested experiment about investigation of the antibacterial activity of DADSnFeS and compared the difference between DADS-nFeS and DADS.

They clearly discuss their strategy of converting organosulfur compounds to inorganic polysulfides with providing the potential advantages for practical applications. The manuscript in the current form is ready to be accepted.

The only comment is again about Fig.2, this time only part c:

- There is a misprint in Y axis, should be viability;

- According to the legend for Fig. 2c it is about "efficacy of Cys-nFeS on the amount of cysteine input to the solvothermal synthesis" but the bars are described rather as organic Cys at different concentrations. This has to be corrected.

REVIEWERS' COMMENTS:

Reviewer #1 (Remarks to the Author):

The authors carefully considered the comments and performed the requested additional experiments in the revised version of the manuscript. I would publish this paper.

Response: We appreciate the positive recommendation for publication of our paper.

Reviewer #2 (Remarks to the Author):

While the authors have adequately addressed the technical concerns, the manuscript requires careful editing to improve the language and clarity and correct numerous grammatical errors. The authors should use the services of an editor expert in English to make the manuscript acceptable for publication. A few, but not all of the problems are noted below.

Response: We appreciate that the reviewer carefully checked our manuscript and pointed out the language issues. We have used an English editing service to improve the language in the revised manuscript.

Line 67 (while "a" small number ...)

Response: We have revised it.

Lines 74, 75

Response: We have revised it.

Line 88 "sulfur element"??? do you mean "elemental sulfur"

Response: Yes, we have corrected it in the revised manuscript.

Line 98 -- do not capitalize "selected-area electron diffraction"

Response: We have revised it.

Lines 114-116 "kcal"; "formed" doesn't make sense; meaning of pairs of numbers "(69.358/50.69) and (66.90/70.69) is unclear -- why pairs of numbers??

Response: The pairs of numbers are given for the two C-S bonds in DAS or methionine. We have corrected it with "kcal" and deleted "formed" in the revised manuscript.

Lines 123-130 -- many mistakes of grammar, etc. Thorough editing needed here.

Response: We have requested an English Editor to check and revise the language for our paper.

Line 159 "crystalline"

Response: We have revised it.

Line 187-188 poor English – rewrite

Response: We have revised it.

Line 224 doesn't make sense

Response: We have revised it.

Line 258 Poor English -- rewrite ("Regarding to the components")

Response: We have revised it.

Line 429 doesn't make sense ("Inconsistence")

Response: We have revised it.

Line 441 Poor English "But..."

Response: We have revised it.

Line 451 Poor English ("releasing mode of polysulfanes")

Response: We have revised it.

Line 462 Poor English "destructured"

Response: We have revised it.

Main References and Supplementary References: proper journal style should be followed throughout using abbreviations and capitalized first letters for journal names,

Response: We have corrected the style for all the references in the revised manuscript.

Reviewer #3 (Remarks to the Author):

The authors have significantly improved the quality of written manuscript by providing all necessary and previously missing references, paying more attention to the details such providing concentrations and statistical analyses. They also included additional and requested experiment about investigation of the antibacterial activity of DADSnFeS and compared the difference between DADS-nFeS and DADS.

They clearly discuss their strategy of converting organosulfur compounds to inorganic polysulfides with providing the potential advantages for practical applications. The manuscript in the current form is ready to be accepted.

Response: We appreciate the positive comments and the recommendation of acceptance for our work.

The only comment is again about Fig.2, this time only part c:

- There is a misprint in Y axis, should be viability;

Response: We have corrected it.

- According to the legend for Fig. 2c it is about "efficacy of Cys-nFeS on the amount of cysteine input to the solvothermal synthesis" but the bars are described rather as organic Cys at different

concentrations. This has to be corrected.

Response: Thanks the reviewer for pointing out this problem. We have corrected it in the revised manuscript.